# FREE LUNCH ALIGNMENT OF TEXT-TO-IMAGE DIFFUSION MODELS WITHOUT PREFERENCE IMAGE PAIRS

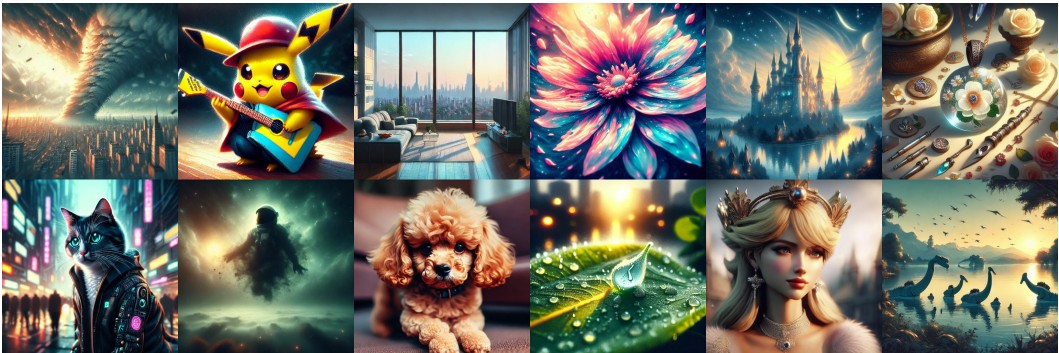

Figure 1: Image generated by our aligned StableDiffusion 1.5 model. Notably, our model is trained on "free lunch" text preference data and does not require access to human preference data.

## ABSTRACT

Recent advances in diffusion-based text-to-image (T2I) models have led to remarkable success in generating high-quality images from textual prompts. However, ensuring accurate alignment between the text and the generated image remains a significant challenge for state-of-the-art diffusion models. To address this, existing studies often employ reinforcement learning with human feedback (RLHF) to align T2I outputs with human preferences. These methods, however, either rely directly on paired image preference data or require a learned reward function, both of which depend heavily on costly, high-quality human annotations and thus face scalability limitations. In this work, we introduce **Text Preference Optimization (TPO)**, a novel framework that enables "free-lunc" alignment of T2I models, achieving alignment without the need for paired image preference data. TPO works by training the model to prefer matched prompts over mismatched prompts, which are constructed by perturbing original captions using a large language model (LLM). Our framework is general and compatible with existing preference-based algorithms. We extend both DPO and KTO to our setting, resulting in **TDPO** and **TKTO**. Quantitative and qualitative evaluations across multiple benchmarks show that our methods consistently outperform their original counterparts, yielding superior human preference scores and better text-to-image alignment.

## 1 INTRODUCTION

Text-to-Image (T2I) models, driven by diffusion(Karras et al., 2022; Kingma et al., 2023; Rombach et al., 2022a; Song et al., 2020), rectified flow (Lipman et al., 2022; Liu et al., 2022), and next-token prediction methods (Wang et al., 2024), have seen significant advances in generating high-quality images from textual descriptions. This is achieved by pretraining on large-scale, high-quality image caption datasets. However, the trained model's performance relies heavily on the quality of the pretraining datasets, and does not necessarily reflect human preferences for image quality and textual alignment. To mitigate this, recent studies have focused on aligning text-to-image (T2I) models with human preferences in the post-training phase.

Motivated by the success of large language models (LLMs), a line of work has introduced reinforcement learning from human feedback (RLHF) for T2I alignment (Black et al., 2024; Clark et al., 2024; Fan et al., 2023; Lee et al., 2023). These approaches involve first training a reward model to predict human preferences and then optimizing the generative model via policy gradient methods. However, this pipeline remains both complex, due to the difficulty of constructing a reliable and stable reward model, and resource-intensive, given the substantial amount of human-labeled data required for effective reward modeling. Moreover, the high-quality image datasets used during initial training offer limited utility at this stage, as alignment with human preferences necessitates a separate and costly collection of preference-specific annotations.

More recently, Direct Preference Optimization (DPO) (Rafailov et al., 2023) and Kahneman–Tversky Optimization (KTO) (Ethayarajh et al., 2024) have simplified this pipeline by formulating preference learning as a single-stage, closed-form optimization problem. These methods have also been extended to align T2I models (Li et al., 2024; Wallace et al., 2024). A significant drawback, however, is their heavy reliance on high-quality, preference-labeled image pairs, which are expensive to collect and have been shown to be vulnerable to noise and inconsistencies in human annotations (Yang et al., 2025). We aim to answer the following question:

*Can we improve text-to-image model alignment without annotating human preference image pairs?*

Motivated by contrastive methods used in training vision-language models, we propose to leverage the high-quality datasets originally used for T2I training by optimizing preference alignment over text pairs rather than image pairs, as in Diffusion-DPO or Diffusion-KTO. This approach is based on the observation that, *given an image-caption dataset, generating mismatched prompts for a given image is significantly easier than constructing image preference pairs for a single prompt*. To enable the construction of such text-preference pairs, we utilize LLMs to generate negative samples—prompts that closely resemble the original but are intentionally mismatched. For example, as shown in Fig. 2, our pipeline flips the word "inside" in the original prompt to "outside", inducing a spatial layout change. This strategy encourages the model to focus on subtle distinctions in prompt semantics, resulting in improved alignment performance. Our contributions are threefold:

- **Preference-data-free alignment.** We propose a novel method for aligning text-to-image diffusion models without requiring human preference data, offering a "free lunch" post-training solution while improving text-to-image alignment.

- **Generalizable pipeline.** Our approach is model-agnostic and can be seamlessly integrated into any RLHF-based method that utilizes preference pairs, making it broadly applicable across existing alignment frameworks.

- **State-of-the-art results.** By adapting DPO and KTO into our framework—TDPO and TKTO—we achieve state-of-the-art performance both qualitatively and quantitatively, surpassing the baselines without using any human preference annotations.

## 2 RELATED WORKS

**Text-to-Image Models.** Text-to-image diffusion models have emerged as one of the most powerful and widely adopted generative techniques for image synthesis (Balaji et al., 2022; Chang et al., 2023; Ho et al., 2020; Karras et al., 2022; Kingma et al., 2023; Rombach et al., 2022a; Shi et al., 2020; Song et al., 2020). These models are capable of generating high-quality images that closely match the semantics of a given text prompt. Despite their impressive performance, achieving precise alignment between textual descriptions and visual outputs remains a key challenge. Recent works have explored improvements through various directions, such as enhanced text encoding (Liu et al., 2024b; Ma et al., 2024; Wu et al., 2023a), improved text embedding interaction architectures (Esser et al., 2024; Liu et al., 2024a), better image captioning (Betker et al., 2023; Lei et al., 2023), and inference-time strategies (Jiang et al., 2024; Prabhudesai et al., 2024; Shen et al., 2024; Wallace et al., 2023). Orthogonal to these approaches, our work focuses on improving text-image alignment through a novel preference optimization strategy, providing a complementary direction to existing methods.

**LLM Alignment.** In recent years, large language models (LLMs) have grown rapidly in scale and capability, demonstrating impressive generative performance across a wide range of language tasks. This power, however, comes with the risk of harmful or undesirable behavior. To mitigate such risks, Reinforcement Learning from Human Feedback (RLHF) (Christiano et al., 2023; Ouyang et al., 2022)

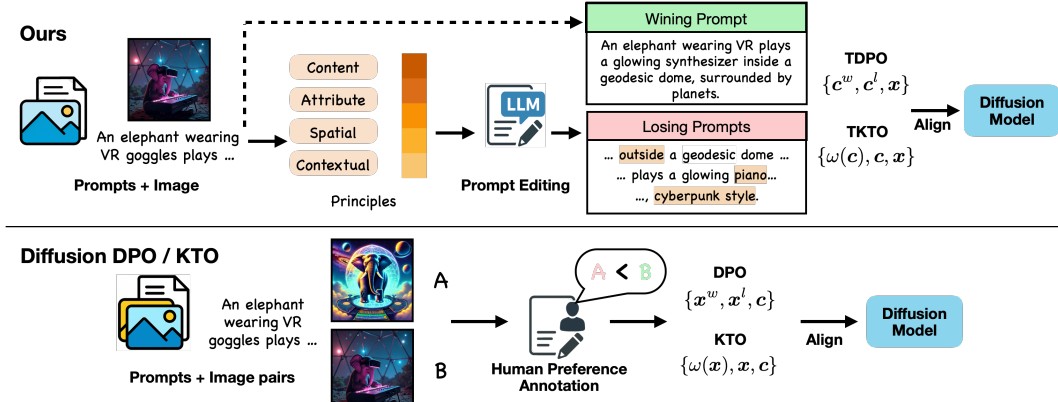

Figure 2: Overview of our Text Preference Optimization (TPO) alignment framework versus the standard Diffusion-DPO/KTO pipeline. **(Top)** We leverage LLMs to perform prompt editing under four principles (content, attribute, spatial, contextual), automatically generating mismatched prompts to form winning/losing text pairs. These prompt pairs are then used to align the diffusion model via our TDPO and TKTO variants in a "free lunch" manner. **(Bottom)** In contrast, existing Diffusion-DPO/KTO methods rely on costly human-annotated image preference pairs.

was introduced. In RLHF, human annotators rank model outputs to create a preference dataset; a reward model is then trained to predict these human preferences, and the LLM is fine-tuned to maximize the learned reward. More recently, alignment methods that avoid an explicit reward-modeling stage have emerged. Direct Preference Optimization (DPO) (Rafailov et al., 2023) represents preferences implicitly via a Bradley–Terry model, allowing the policy to be optimized in closed form. Several variants (Ethayarajh et al., 2024; Shao et al., 2024; Wu et al., 2024) and extensions of DPO have further simplified and improved LLM alignment. While these techniques were first developed for LLMs, our focus is on adapting these concepts to align text-to-image diffusion models. We build on the optimization strategies pioneered in language alignment and adapt them to the multimodal setting.

**Text-to-Image Preference Optimization.** As with LLMs, text-to-image (T2I) diffusion models must also be aligned to respect user preferences and safety constraints. Building on the success of preference-based methods such as DPO, several studies have recently transplanted these objectives to the T2I setting and reported strong gains (Karthik et al., 2024; Li et al., 2024; Miao et al., 2024; Wallace et al., 2024; Yang et al., 2024). In particular, *DiffusionDPO* (Wallace et al., 2024) and *DiffusionKTO* (Li et al., 2024) extend the DPO and KTO objectives (Ethayarajh et al., 2024) to diffusion-based generators. DSPO (Zhu et al., 2025) aims to close the gap between preference alignment method used in LLM and T2I diffusion models by leveraging score matching. We introduce a unified framework that is agnostic to the specific preference-alignment loss and can embed any diffusion T2I alignment objective. To illustrate its flexibility, we show how both DiffusionDPO and DiffusionKTO instantiate naturally within our formulation.

## 3 METHOD

The effectiveness of preference alignment methods for diffusion models (Wallace et al., 2024; Li et al., 2024) depends heavily on access to high-quality image preference datasets. However, collecting such datasets at scale is challenging due to several key limitations. First, collecting human preferences is expensive, as it requires substantial manual effort for both annotation and validation. Moreover, when the underlying diffusion model changes (*e.g.*, from Stable Diffusion 1.5 to 3.0), previously collected preference data may no longer be effective, requiring a fresh round of data collection. Second, human preferences reflect a mixture of factors, including image quality, alignment with the text prompt, and subjective aesthetic judgment, making them inherently noisy and inconsistent.

Our motivation stems from the observation that generating matched and mismatched text pairs is significantly easier than collecting image preference pairs. By leveraging LLMs to generate text preference pairs for each image, we can align text-to-image models with minimal human supervision, offering a highly cost-effective alternative to manual preference annotation. This enables scalable

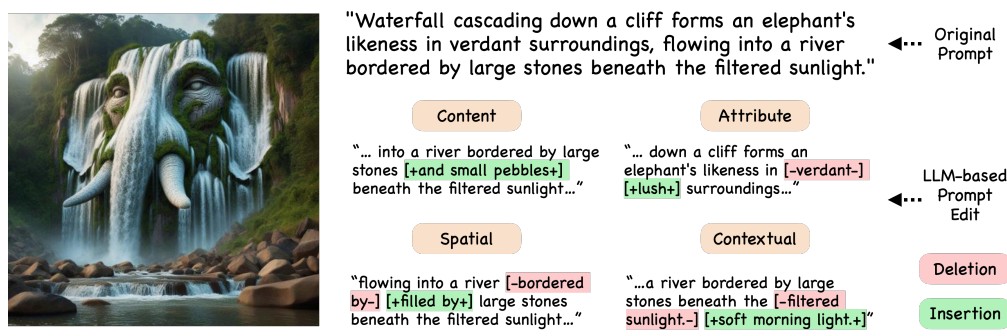

Figure 3: An example of how the four modification principles (content, attribute, spatial, contextual) are applied on a given image-prompt pair.

and efficient alignment at virtually no additional labeling cost. More formally, given a paired dataset $\{x, c^w, c^l\}$ where $x$ is an high-quality image from a human-curated dataset, $c^w$ and $c^l$ are the matched and mismatched captions, respectively, we aim to learn a new model $p_\theta(x|c)$ that can achieve better on both text-to-image alignment and human preference alignment. Below, we first introduce our text preference pair construction pipeline, then formally derive our method.

### 3.1 TEXT PREFERENCE CONSTRUCTION WITH LLMS

Given a paired dataset $\{x, c\}$, where $x$ is an image and $c$ is its corresponding caption, we aim to construct a new dataset with triplets $x, c^w, c^l$, where $c^w$ aligns better with the image $x$ than $c^l$. We assume the original dataset is of high quality, meaning that the caption $c$ accurately describes the image. Based on this assumption, we use LLMs to generate negative captions $c^l$ while using the original caption from the dataset as $c^w = c$. To achieve this, we prompt LLMs to modify the ground truth caption such that the perturbed caption describes an image that is visually distinct from the original. More specifically, we define the four core principles for LLMs to follow when editing the prompt:

- **Content Modifications**: Altering the presence or quantity of concrete objects in the scene. This includes adding or removing objects, replacing one object with another, or changing the number of instances (*e.g.*, modifying "three trees" to "five trees").
- **Attribute and Descriptor Modifications**: Changing visual or stylistic properties of objects, such as material, texture, or style. This also includes enriching descriptions with detailed qualifiers, while avoiding trivial edits like simple color changes.
- **Spatial and Dynamic Modifications**: Modifying the spatial arrangement or motion state of objects. Examples include adjusting object poses, changing spatial relations (foreground/background), or altering object alignment and composition.
- **Contextual and Environmental Modifications**: Editing elements related to the broader scene context, such as background, weather, lighting, or time of day. Changes may also involve shifting the physical or cultural setting (*e.g.*, urban vs. historical).

An example of how these four principles are applied is shown in Fig. 3. More examples and complete prompts for these principles are provided in Appendix A.4 and Appendix A.5.

**Preference Optimization over Input Conditions.** In our setting, given an image $x$, we have a matching text prompt $c^w$ and a set of mismatched prompts $\{c^l_i\}_{i=1}^N$. Unlike standard DPO and KTO, where preference optimization is performed over output images, our goal is to align the model with both matched and mismatched input conditions. In this section, we derive the text preference optimization objectives for DPO and KTO under our setting, which we refer to as Text Preference DPO (TDPO) and Text Preference KTO (TKTO), respectively.

**TDPO.** For Text Preference DPO, we have a triplet $\{c^w, c^l, x\}$ for each image. We have access to the preference relation $c^w \succ c^l \mid x$, where $c^w$ and $c^l \in \{c^l_i\}_{i=1}^N$ denote the matched and mismatched text prompts, respectively. Following Rafailov et al. (2023); Wallace et al. (2024), we use the

Bradley-Terry (BT) (Bradley & Terry, 1952) model to model this preference as

$$p_{\mathrm{BT}}(\boldsymbol{c}^w \succ \boldsymbol{c}^l \mid \boldsymbol{x}) = \sigma(r(\boldsymbol{c}^w, \boldsymbol{x}) - r(\boldsymbol{c}^l, \boldsymbol{x})), \tag{1}$$

where $\sigma$ is the sigmoid function and the reward model $r_\theta$ is a neural network parameterized by $\theta$. Our training objective is to minimize the negative log-likelihood of this preference:

$$L = -\mathbb{E}_{\boldsymbol{c}^w, \boldsymbol{c}^l, \boldsymbol{x}}[\log \sigma(r(\boldsymbol{c}^w, \boldsymbol{x}) - r(\boldsymbol{c}^l, \boldsymbol{x}))] \tag{2}$$

To simplify this optimization and avoid explicit reward modeling, DPO implicitly represents the reward function. Following the derivations in Rafailov et al. (2023), the reward function can be expressed by optimal $p_\theta^*$ and $p_{\mathrm{ref}}$, and by taking Bayes' rule, we get:

$$r(\boldsymbol{c}, \boldsymbol{x}) = \beta \log \frac{p_\theta^*(\boldsymbol{c}|\boldsymbol{x})}{p_{\mathrm{ref}}(\boldsymbol{c}|\boldsymbol{x})} + \beta \log Z(\boldsymbol{x}) = \beta \left[ \log \frac{p_\theta^*(\boldsymbol{x}|\boldsymbol{c})}{p_{\mathrm{ref}}(\boldsymbol{x}|\boldsymbol{c})} - \log \frac{p_\theta^*(\boldsymbol{x})}{p_{\mathrm{ref}}(\boldsymbol{x})} \right] + \beta \log Z(\boldsymbol{x}) \tag{3}$$

Here in the second equality of Eq. (3), we assume that the text condition is sampled from the dataset $c \sim D_c$ and is independent of the model parameters $\theta$. Consequently, we have $p^\theta(c) = p\mathrm{ref}(c)$. However, $p_\theta(\boldsymbol{x}) \neq p_{\mathrm{ref}}(\boldsymbol{x})$ since the distribution of the generated image is characterized by $\theta$. We can then substitute Eq. (3) into Eq. (2) and get:

$$L_{\mathrm{TDPO}}(\theta) = -\mathbb{E}_{\boldsymbol{c}^w, \boldsymbol{c}^l, \boldsymbol{x}} \left[ \log \sigma \left( \beta \log \frac{p_\theta(\boldsymbol{x}|\boldsymbol{c}^w)}{p_{\mathrm{ref}}(\boldsymbol{x}|\boldsymbol{c}^w)} - \beta \log \frac{p_\theta(\boldsymbol{x}|\boldsymbol{c}^l)}{p_{\mathrm{ref}}(\boldsymbol{x}|\boldsymbol{c}^l)} \right) \right] \tag{4}$$

where optimizing Eq. (4) is equivalent to maximizing Eq. (1).

**TKTO.** For Text Preference KTO, we have input $\{\boldsymbol{c}, \boldsymbol{x}, \omega(\boldsymbol{c})\}$ where $\omega(\boldsymbol{c}) = 1$ if $\boldsymbol{c} = \boldsymbol{c}^w$ and $\omega(\boldsymbol{c}) = -1$ for $\boldsymbol{c} \in \{\boldsymbol{c}_i^l\}_{i=1}^N$. KTO seeks to maximize the following objective:

$$\max_r \quad \mathbb{E}_{\boldsymbol{c}, \boldsymbol{x}}[U(\omega(\boldsymbol{c})(r(\boldsymbol{c}, \boldsymbol{x}) - z_0))] \tag{5}$$

Here $U$ is the utility function, where $r(\boldsymbol{c}, \boldsymbol{x})$ and $z_0$ are defined as:

$$r(\boldsymbol{c}, \boldsymbol{x}) = \beta \log \frac{p_\theta(\boldsymbol{x}|\boldsymbol{c})}{p_{\mathrm{ref}}(\boldsymbol{x}|\boldsymbol{c})}; \quad z_0 = \mathrm{sg}\left[ \beta KL(p_\theta(\boldsymbol{x}|\boldsymbol{c})||p_{\mathrm{ref}}(\boldsymbol{x}|\boldsymbol{c})) \right] \tag{6}$$

where sg refers to the stop gradient operator. Following Kahneman-Tversky's prospect theory (Tversky & Kahneman, 1992) , we use a centered sigmoid function as utility function. This gives us the training objective for TKTO:

$$L_{\mathrm{TKTO}}(\theta) = -\mathbb{E}_{\boldsymbol{c}, \boldsymbol{x}} \left[ -\sigma(\omega(\boldsymbol{c})\beta(\log \frac{p_\theta(\boldsymbol{x}|\boldsymbol{c})}{p_{\mathrm{ref}}(\boldsymbol{x}|\boldsymbol{c})} - z_0)) \right] \tag{7}$$

### 3.2 TEXT PREFERENCE OPTIMIZATION FOR DIFFUSION MODEL

We now extend our TPO alignment algorithm to align diffusion-based T2I models.

**Diffusion Model.** Diffusion models (Song et al., 2020; Ho et al., 2020; Kingma et al., 2023) are generative models that sample from a learned distribution $p_\theta(\boldsymbol{x}_0)$, trained to approximate the empirical data distribution $q(\boldsymbol{x}_0)$. Training proceeds by learning to invert a fixed forward process of (denoising) diffusion $q(\boldsymbol{x}_t|\boldsymbol{x}_{t-1})$. The forward process $q(\boldsymbol{x}_{1:T} \mid \boldsymbol{x}_0)$ is a Markov chain with Gaussian transition probabilities governed by noise schedules $\{\alpha_t, \sigma_t\}$, as defined in Rombach et al. (2022a), that gradually add noise to data $\boldsymbol{x}_0$. The reverse (denoising) process is defined by

$$p_\theta(\boldsymbol{x}_{0:T}) = p(\boldsymbol{x}_T) \prod_{t=1}^T p_\theta(\boldsymbol{x}_{t-1} \mid \boldsymbol{x}_t), \quad \text{where } p_\theta(\boldsymbol{x}_{t-1} \mid \boldsymbol{x}_t) = \mathcal{N}\left(\boldsymbol{x}_{t-1}; \mu_\theta(\boldsymbol{x}_t, t), \tilde{\sigma}_t^2 I\right). \tag{8}$$

A neural network $\epsilon_\theta(\boldsymbol{x}_t, t)$ is trained to predict the noise $\epsilon$ in $\boldsymbol{x}_t$ by minimizing the simplified evidence lower bound associated to this model,

$$L_{\mathrm{DM}} = \mathbb{E}_{\boldsymbol{x}_0 \sim q(\boldsymbol{x}), t \sim \mathcal{U}[0,T], \boldsymbol{\epsilon} \sim \mathcal{N}(0,I), \boldsymbol{x}_t \sim q(\boldsymbol{x}_t|\boldsymbol{x}_0)} \left[ w(t) \left\| \boldsymbol{\epsilon} - \boldsymbol{\epsilon}_\theta(\boldsymbol{x}_t, t) \right\|_2^2 \right], \tag{9}$$

where $w(t)$ is a weighting function.

**Diffusion TDPO.** Existing work has adapted DPO into the field of diffusion model (Wallace et al., 2024; Yang et al., 2024). Diffusion-DPO (Wallace et al., 2024) frames the diffusion process as a MDPs and defines the reward function as the reward of the whole chain, $r(\boldsymbol{c}, \boldsymbol{x}_0) = \mathbb{E}_{p_\theta(\boldsymbol{x}_{1:T}|\boldsymbol{x}_0,\boldsymbol{c})}[R(\boldsymbol{c}, \boldsymbol{x}_{0:T})]$. It then utilizes an evidence lower bound to adapt the training objective of DPO into the diffusion model setting. For the detail of this adaptation, please look at section 4 of Diffusion-DPO (Wallace et al., 2024). Following their work and Eq. (4), we can similarly derive the naive DiffusionTDPO objective, it can be rewritten as:

$$L_{\text{Diff-TDPO}} = -\mathbb{E}_{\boldsymbol{x}_0, \boldsymbol{c}^w, \boldsymbol{c}^l, t, \boldsymbol{x}_t}[\log \sigma(-\beta w(t)(||\boldsymbol{\epsilon}_\theta(\boldsymbol{x}_t, \boldsymbol{c}^w, t) - \boldsymbol{\epsilon}^w||_2^2 - ||\boldsymbol{\epsilon}_{\text{ref}}(\boldsymbol{x}_t, \boldsymbol{c}^w, t) - \boldsymbol{\epsilon}^w||_2^2$$
$$- (||\boldsymbol{\epsilon}_\theta(\boldsymbol{x}_t, \boldsymbol{c}^l, t) - \boldsymbol{\epsilon}^l||_2^2 - ||\boldsymbol{\epsilon}_{\text{ref}}(\boldsymbol{x}_t, \boldsymbol{c}^l, t) - \boldsymbol{\epsilon}^l||_2^2)))], \quad (10)$$

where $\boldsymbol{\epsilon}_\theta$ and $\boldsymbol{\epsilon}_{\text{ref}}$ are our training model and pretrained frozen model, $\mathcal{U}$ is a uniform distribution.

**Diffusion TKTO.** With the success of previous work of adapting DPO into the diffusion model, another alignment optimization algorithm , KTO, has also been adapted to the this field. Based on the work of Diffusion-KTO (Li et al., 2024) and Eq. (11), we present the Diffusion TKTO as:

$$L_{\text{Diff-TKTO}} = -\mathbb{E}_{\boldsymbol{x}_0, \boldsymbol{c}, t, \boldsymbol{x}_t}\sigma(w(\boldsymbol{c})\beta\left[-(||\boldsymbol{\epsilon} - \boldsymbol{\epsilon}_\theta(\boldsymbol{x}_t, t, \boldsymbol{c})||_2^2 - ||\boldsymbol{\epsilon} - \boldsymbol{\epsilon}_{\text{ref}}(\boldsymbol{x}_t, t, \boldsymbol{c})||_2^2)-z_0\right]) \quad (11)$$

here $w(\boldsymbol{c}) = \pm 1$ if the prompt $\boldsymbol{c}_0$ is matched or mismatched. Consistent with Li et al. (2024), we set $z_0 = \text{sg}\left[\beta KL(p_\theta(\boldsymbol{x}|\boldsymbol{c})||p_{\text{ref}}(\boldsymbol{x}|\boldsymbol{c}))\right]$. In practice, we use a biased but low-variance estimator for $z_0$:

$$\max\left(0, \frac{1}{m}\sum \beta(-(||\boldsymbol{\epsilon} - \boldsymbol{\epsilon}_\theta(\boldsymbol{x}_t, t, \boldsymbol{c})||_2^2 - ||\boldsymbol{\epsilon} - \boldsymbol{\epsilon}_{\text{ref}}(\boldsymbol{x}_t, t, \boldsymbol{c})||_2^2))\right)$$

## 4 EXPERIMENTS

### 4.1 EXPERIMENTS SETUP

**Datasets and Model.** In this work, all experiments are conducted using Stable Diffusion v1.5 (SD v1.5) (Rombach et al., 2022b). We fine-tune SD v1.5 on the Human Preference Synthetic Dataset (HPSD) (Egan et al., 2024), which comprises one million high-quality image–caption pairs. For evaluation, we follow previous work (Wallace et al., 2024; Li et al., 2024) and use the same HPDv2 test set (Wu et al., 2023b), Pick-a-Pic v2 test dataset (Kirstain et al., 2023) and Parti-Prompts dataset (Yu et al., 2022) for evaluation. In addition, we also include open-image-preferences-v1 (Berenstein et al., 2024) dataset for evaluation. This dataset contains prompts collect from everyday image-generation user requests from online platforms and have been filtered through a toxicity-reduction pipeline, making it an especially robust evaluation suite. Details on each dataset are provided in the Appendix A.2.

**Training Setup and Baselines.** Our training involves two stages, both of which fine-tune only the U-Net of SD v1.5 while freezing all other components. In the first stage, we fine-tune the pretrained SD v1.5 model on the HPSD dataset with a pure SFT loss objective until we observe convergence. This stage is to close the gap between the pretrained model and the target dataset, such SFT stage is common in RLHF fine-tuning in LLM literature (Christiano et al., 2023; Ouyang et al., 2022). In Stage 2, we continue fine-tuning this SFT-adapted model under identical hyperparameters using our TDPO and TKTO methods alongside the Diffusion-DPO and Diffusion-KTO baselines. For Diffusion-KTO and Diffusion-DPO, we need to acquire preference image pairs. Specifically, with the original high-quality image being $\boldsymbol{x}^w$ we need to construct $\boldsymbol{x}^l$. For details of baseline setups, please refer to Sec. 4.1 and Appendix B.7

In our experiments, we observed that direct fine-tuning with the objectives in Eq. (10) and Eq. (11) degraded sample quality after several steps. To address this, we introduce a clipping mechanism inspired by Proximal Policy Optimization (Schulman et al., 2017) for more stable training. Concretely, we clamp the squared L2 norm of the negative-sample term in the loss, which bounds extreme negative signals and stabilizes training (see Appendix B.1 for details).

**Evaluation.** In our evaluation, we generate images from each model using the same test prompts and assess alignment with five metrics: PickScore (Kirstain et al., 2023), CLIP alignment (Radford et al., 2021), HPSv2 (Wu et al., 2023b), and ImageReward (Xu et al., 2023). For each metric, we compute the win rate against the SD v1.5 baseline i.e., the fraction of prompts on which a model's score exceeds SD v1.5's, and report the average win rate across all five metrics. We also present the mean metric scores for each model in the Appendix B.2. All evaluations use identical sampling settings (fixed random seed, classifier-free guidance scale of 7.5, and 50 diffusion steps).

Table 1: Comparison against Diffusion-DPO and Diffusion-KTO Baselines on different testsets. Win rates (%) for two sectors (each merges two datasets). Best in **bold**, second in underline. Our methods shaded in blue. IR = Image Reward.

| Method | HPSv2 | | | | PartyPrompt | | | |
|---|---|---|---|---|---|---|---|---|
| | **PS** | **CLIP** | **HPS** | **IR** | **PS** | **CLIP** | **HPS** | **IR** |
| SFT | 76.25 | 52.00 | 75.75 | 76.00 | 69.06 | 52.38 | 63.31 | 73.81 |
| Diff-DPO | 77.00 | 54.75 | 59.00 | 70.00 | 71.06 | 54.06 | 49.94 | 64.88 |
| Ours-TDPO | **83.25** | **56.00** | 79.00 | **82.25** | **74.63** | 57.25 | 70.19 | **78.75** |
| Diff-KTO | 80.25 | 53.75 | 76.00 | 76.75 | 73.50 | 52.63 | 63.69 | 71.62 |
| Ours-TKTO | 80.00 | 55.75 | **81.00** | 80.75 | 73.62 | **57.31** | **70.63** | 77.31 |
| | Pick-A-Pic | | | | OpenImagePref | | | |
| SFT | 71.20 | 50.80 | 62.00 | 72.80 | 80.60 | 53.60 | 77.00 | 77.60 |
| Diff-DPO | 72.80 | 54.20 | 52.40 | 68.60 | 83.60 | 57.20 | 62.00 | 73.20 |
| Ours-TDPO | 75.20 | 56.00 | 69.20 | **82.60** | **88.20** | 58.20 | **80.20** | **86.00** |
| Diff-KTO | **77.40** | 55.20 | 65.60 | 74.60 | 82.60 | 57.00 | 76.60 | 80.80 |
| Ours-TKTO | 74.40 | **58.60** | **70.80** | 80.20 | 84.20 | **60.80** | 79.40 | 82.40 |

## 4.2 Results

**Baseline setup.** In Fig. 4, we illustrate several different setups for aligning T2I models: (a) fine-tuning the model directly with desired prompt–image pairs; (b) collecting human preference annotations and aligning with preference-based image pairs; (c) constructing synthetic preference pairs by altering the prompt and generating less-preferred images from the modified prompt; (d) our approach, which directly supervises the model with positive and negative prompt pairs.

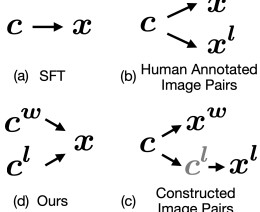

Figure 4: Finetune setup.

**Overall qualitative comparison in win-rate.** Tab. 1 reports the average win rates against SD v1.5 across four datasets and five evaluation metrics. For both Diffusion-DPO and Diffusion-KTO, we adopt setup (c), where the same $c^l$ constructed by our method is used to generate $x^l$ with the supervised fintuned SD v1.5 model. We consider this a fair comparison setup against our approach. Metrics are reported in win rate against sdv1.5. For instance, in Tab. 1, the value 83.25 (at the Ours-TDPO row and at the column of PS under the box of HPSv2) indicates that TDPO outperforms the SDv1.5 83.25% of the time on the HPSv2 dataset, as measured by PickScore. Overall, our methods consistently surpass all baselines, with the exception that Diffusion-KTO achieves slightly higher performance on a few metrics. This demonstrates that our approach achieves stronger alignment under a controlled and comparable evaluation setting. Additional ablations on alternative baselines shown in Fig. 4 are discussed and provided in Appendix B.7 and Tab. 6 of the appendix.

**Comparison with method trained on pick-a-pic preference data.** Tab. 2 Here we compare against methods trained directly on Pick-a-Pic (Kirstain et al., 2023), where human preference annotations over images are available (setting b from Fig. 4). In contrast, our method does not require annotated preference pairs for training; instead, we rely only on captions and the winning images provided by the dataset. We find that while our TDPO consistently outperforms Diffusion-DPO. Our TKTO variant falls short of Diffusion-KTO on HPS and IR scores. This is likely because our approach does not leverage the human-annotated preference information available in the dataset. Nevertheless, our method achieves comparable performance without using any human annotation.

**Qualitative Results.** In Fig. 5, we show a side-by-side comparison of images generated by our method and by the baselines. In the first row, we can see that only our methods, TDPO and TKTO, render both "twilight" and "misty" distinctly. Other baselines struggle to convey either concept. In the second row, all methods except vanilla SDv1.5 hint at something "behind" the biker, but only ours clearly shows a monster in pursuit. The others look more like a second rider tailing the first. In the third row, our methods and SFT successfully generate a black wolf that is "howling", and only our methods also embed the "forest" setting. The other methods either omit the trees altogether or render them too faintly. In the last row, we can see that the TKTO method precisely generates two

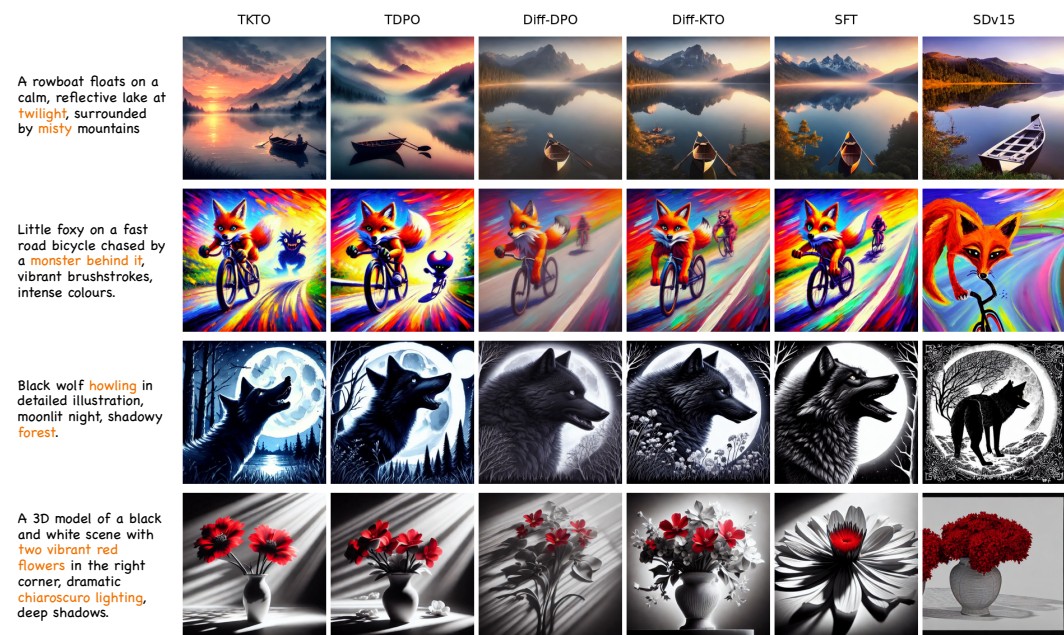

Figure 5: Side-by-Side grid comparison of the image generation using our methods and the baselines. The leftmost column is the prompts used to generate the images. The important concept or element of the prompt that the model successfully or fails to capture is printed in orange color.

Table 2: Comparison to previous methods trained on the pick-a-pick human preference dataset with Win rates (%) against SD-1.5 are reported. Our method only uses the winning image $x^w$. Metrics with * are directly taken from Zhu et al. (2025).

| Method | Supervision | PICK-A-PIC | | | | PARTI-PROMPT | | | |
|---|---|---|---|---|---|---|---|---|---|
| | | PS | CLIP | HPS | IR | PS | CLIP | HPS | IR |
| SFT* | $(x, c)$ | 70.20 | 61.20 | 84.20 | 76.40 | 64.27 | 54.72 | 85.72 | 71.38 |
| Diff.-DPO* | $(x^w, x^l, c)$ | 71.60 | 58.80 | 70.20 | 63.60 | 61.18 | 55.45 | 66.48 | 62.19 |
| Ours-TDPO | $(x, c^w, c^l)$ | 70.60 | 62.60 | 75.80 | 75.80 | **67.12** | 57.06 | 69.88 | 67.75 |
| Diff.-KTO* | $(x^w, x^l, c)$ | 71.40 | 60.02 | 84.40 | 77.00 | 64.80 | 54.34 | 86.16 | 71.51 |
| Ours-TKTO | $(x, c^w, c^l)$ | **74.60** | **63.80** | 71.40 | 74.00 | 65.25 | **59.63** | 68.88 | 67.75 |
| DSPO* | $(x^w, x^l, c)$ | 73.60 | 61.80 | **84.80** | **78.00** | 65.32 | 54.86 | **87.50** | **71.75** |

red flowers against a dark background with strong, directional light and deep shadows. The baselines produce the wrong number of red flowers and offer only weak chiaroscuro effects. Overall, these examples demonstrate that TKTO consistently captures each prompt's detailed concepts and produces images that align more faithfully with what the user asked for.

**Result on SDXL** Placeholder here for SDXL.

**Quantitative results from Human Studies**

We conduct a human study comparing DiffusionDPO, supervised fine-tuning (SFT), and our TDPO method, focusing on prompt–image alignment. For DiffusionDPO, we use the original checkpoint provided in Wallace et al. (2024), trained on the Pick-a-Pic (Kirstain et al., 2023) dataset. For TDPO, we adopt the Table 2 setting, where only the winning images from Pick-a-Pic are used. For the SFT baseline, we fine-tune SD1.5 on the Pick-a-Pic winning images. This setup ensures a fair comparison, as all methods are trained on the same set of winning image–prompt pairs. While DiffusionDPO additionally has access to human-dispreferred images, TDPO instead leverages generated negative prompts. More details of how human studies are collected can be found in Appendix A.7.

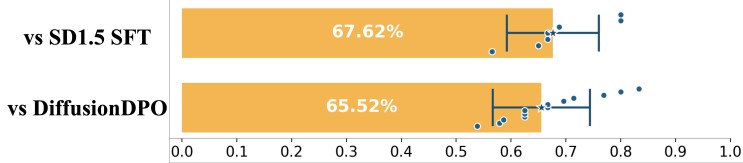

Figure 6: Human studies on Prompt to Image Alignment. Top Our TDPO vs SD1.5 SFT, bottom TDPO vs DiffusionDPO.

Table 3: Ablation on prompt editing principles. Left: TDPO. Right: TKTO. Metrics are reported as win rates (%).

| | TDPO | | | | | TKTO | | | |
|---|---|---|---|---|---|---|---|---|---|
| Model | PS | CLIP | HPS | IR | Model | PS | CLIP | HPS | IR |
| TDPO | 74.6 | 57.3 | 70.2 | 78.8 | TKTO | 73.6 | 57.3 | 70.6 | 77.3 |
| w/o Attribute | 74.3 | 56.3 | 71.6 | 78.3 | w/o Attribute | 72.4 | 57.4 | 70.1 | 78.2 |
| w/o Content | 75.2 | 55.6 | 67.4 | 76.1 | w/o Content | 74.9 | 56.1 | 69 | 76.7 |
| w/o Contextual | 73.6 | 57.1 | 70.8 | 77.0 | w/o Contextual | 69.2 | 56.1 | 67.6 | 73.8 |
| w/o Spatial | 76.4 | 56.4 | 70.4 | 77.2 | w/o Spatial | 74.4 | 57 | 72.8 | 78 |

### 4.3 ABLATIONS AND DISCUSSIONS

**Implicit preference over text prompts correlates with human preferences over images.** *How would learning preferences over text condition improve human preferences?* Here we shed a light on this matter in Fig. 7. To begin with, we define a metric called implicit preference score(IPS), which shows how much a model prefers $x, c^w$ over $x, c^l$ as follows:

$$\mathbf{IPS} = \mathbb{E}_{t\sim\mathcal{U},\boldsymbol{x}_t\sim q(\boldsymbol{x}_t|\boldsymbol{x}_0)}[\|\epsilon - \epsilon_\theta(\mathbf{x}_t, t, \boldsymbol{c}^l)\|_2^2 - \|\epsilon - \epsilon_\theta(\mathbf{x}_t, t, \boldsymbol{c}^w)\|_2^2]. \quad (12)$$

The term, intuitively interpretable as the difference in diffusion loss between negative and positive pairs, quantifies how much more likely the model is to generate image $x$ given the matching prompt $c^w$ compared to the mismatched prompt $c^l$. A larger value indicates better alignment with the ground truth textual preference. The regression plots in Fig. 7 reveal a clear positive correlation between human preference metrics and implicit preference score: models with higher preference score exhibit higher human preference metrics. *This suggests an underlying connection between alignment with textual prompt pairs and alignment with human image preferences.* Our methods, including their TDPO and TKTO variants, appear in the top-right region of the plots, indicating both the highest implicit preference score and the highest human preference scores among almost all evaluated baselines.

**The effectiveness of prompt editing principles.** Despite the excellent performance improvement shown in Tab. 1, we are yet to answer the question: *What is the benefit of the improvement brought by each editing principle?* To this end, we conduct an ablation study of the effect of how each of the editing principles defined in Sec. 3.1 affect the performance of alignment. Specifically, in this ablation study, we fine-tune with four different settings, in each setting, one of the editing principles is dropped. Tab. 3 shows that dropping different modification principles leads to different performance among the metrics and methods. Interestingly, we observe that removing content-related modifications leads to a significant drop in CLIP score. This is likely because content modifications most directly influence the model's sensitivity to the semantic meaning of prompts. We also observe that dropping spatial-related modification leads to tradeoff, and even improvement for TKTO. The plausible cause is that spatial prompts inject high-variance, low-signal supervision: the viewpoint of the image is often underspecified, which bring ambiguity to spatial predicates like "left" and "right", and enforcing spatial changes entangles cross-attention with object layout, which destabilizes the overall learning.

## 5 CONCLUSION

We have presented a novel "free lunch" alignment method that leverages LLM-generated text–preference pairs to fine-tune text-to-image diffusion models without requiring human pref-

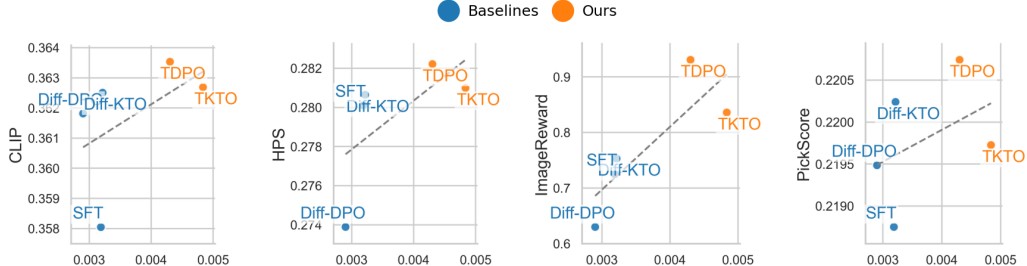

Figure 7: Regression plots between alignment metrics and implicit preference score. These plots shows a positive correlation between alignment metrics and implicit preference score. Our method consistently achieves higher implicit preference score and higher human preference scores.

erence annotations. Our instantiations, TDPO and TKTO, achieve consistent improvements over the baselines, while remaining model-agnostic and easily integrated into any RLHF-style pipeline. Future work includes extending this framework by integrating other preference-optimization algorithms, applying it to other modalities such as text-to-video and text-to-3D generation, and exploring richer negative-sample generation techniques for greater diversity.

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

# A   MORE EXPERIMENT DETAILS

## A.1   IMPLICIT PREFERENCE SCORE

Recall that in Sec. 4.3 we defined the implicit preference score for a triplet $(\boldsymbol{x}, \boldsymbol{c}^w, \boldsymbol{c}^l)$ as

$$\mathbb{E}_{t \sim \mathcal{U}, \boldsymbol{x}_t \sim q(\boldsymbol{x}_t|\boldsymbol{x}_0)}[\|\boldsymbol{\epsilon} - \boldsymbol{\epsilon}_\theta(\boldsymbol{x}_t, t, \boldsymbol{c}^l)\|_2^2 - \|\boldsymbol{\epsilon} - \boldsymbol{\epsilon}_\theta(\boldsymbol{x}_t, t, \boldsymbol{c}^w)\|_2^2]. \tag{13}$$

In practice, we fix the diffusion timestep to $t = 0.5$, sample $\boldsymbol{x}_t$ three times for each triplet, and average the resulting diffusion losses to compute implicit preference score. All triplets are drawn from the HPSD evaluation set. The mismatched prompt $\boldsymbol{c}^l$ is generated by applying a single modification to the original prompt, as described in Sec. 4.3.

We observe a strong negative correlation between implicit preference score and human preference metrics. In other words, models that incur higher diffusion loss on the mismatched prompt $\boldsymbol{c}^l$ relative to the matched prompt $\boldsymbol{c}^w$ consistently receive higher human preference scores. This result aligns with our goal of improving text–image alignment and helps explain why our method achieves superior performance on human preference evaluations.

## A.2   DATASET

Here, we list all the dataset we have used in this study, with a short introduction and the usage in this sutdy.

- HPDv2 (Wu et al., 2023b) (Apache license 2.0) is a large-scale (798k preference choices / 430k images), a well-annotated dataset of human preference choices on images generated by text-to-image generative models. We have use the prompt of its test set for evaluation, the test set include 400 data.

- Pick-a-Pic v2 (MIT license) (Kirstain et al., 2023) is a large and open dataset for human feedback in text-to-image generation. We use its test set for evaluation, which contain 500 data.

- Parti-prompts (Yu et al., 2022) (Apache license 2.0 license) is a rich set of over 1600 prompts in English that we release as part of this work. We use the whole dataset for evaluation.

- open-image-preferences-v1 (Berenstein et al., 2024) (Apache license 2.0 license) is a dataset contain over 7k human preference pairs on images generated by powerful text-to-image generative models. All the images pairs are generated by the same prompt, and a preference binary label is provided by human annotators. We split the dataset into train and evaluation set, where the the last 500 data are split for evaluation while the rest form the training set.

- HPSD (Egan et al., 2024) (MIT license) dataset comprises of high-quality AI-generated images sourced from various websites and individuals, it contains over 780k image-prompt paris. For the first stage SFT, we use the whole dataset for training. For the second stage of fine-tuning, we use the first 100K.

## A.3   TRAINING DETAILS

All experiments are run on two NVIDIA A100 GPUs using Stable Diffusion v1.5 (CreativeML Open RAIL-M license). Except for SFT fine-tuning, we use a batch size of 16 and a constant learning rate of $1 \times 10^{-6}$. For SFT, we employ a batch size of 256, a learning rate of $1 \times 10^{-5}$, and train for 17 500 steps until a convergence on training loss has been observed. All models are optimized with AdamW ($\beta_1 = 0.9$, $\beta_2 = 0.999$, $\epsilon = 1 \times 10^{-8}$) and a constant learning rate. Our methods (TDPO, TKTO) and the baselines (Diffusion-DPO, Diffusion-KTO) share $\beta = 5000$. To select the best checkpoint, we sample 500 prompts from the HPSD evaluation set, generate corresponding images, and compute four evaluation metrics. We then choose the checkpoint with the highest score across these metrics for final evaluation.

## A.4   PROMPT MODIFICATION EXAMPLES

In this subsection, we provide more examples of prompt modification such as the one in Figure 2, where a given prompt is modified based on our modification principles in Fig. 8.

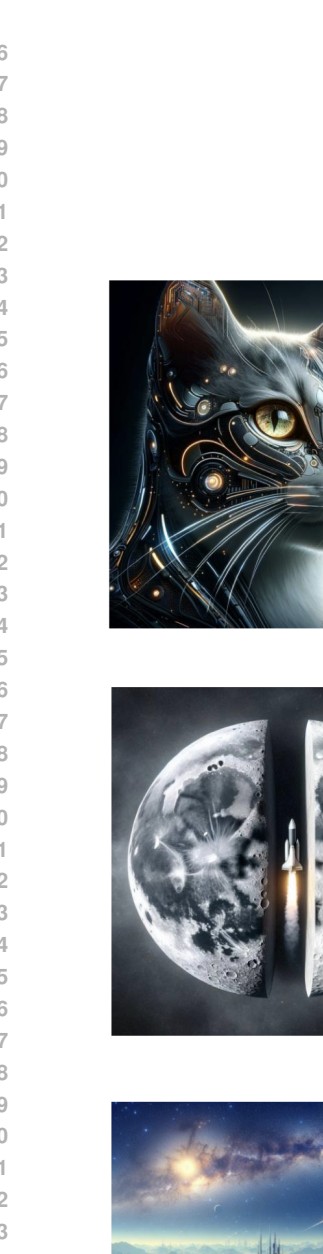

"An artistic rendition of a feline face featuring metallic and electronic accents; its eyes glow golden and emerald, within an intricate design of circuits and neon tones."

**Content**

"... circuits and neon tones, [+with a small antenna on its head+]"

**Attribute**

"... within an [-intricate-] [+cumbersome and complicated+] design of circuits and neon tones."

**Spatial**

"... within an intricate design of circuits and neon tones, [+looking above. +]"

**Contextual**

"... golden and emerald, within an intricate design of circuits [-and neon tones-]."

"An image displays a vertical split moon, one side illuminated showing craters & textures, a dark void separates it from a lit spacecraft launching, set against a starry cosmic background."

**Content**

"... a dark void separates it from a lit [-spacecraft-] [+rocket+] launching, set against .."

**Attribute**

".. one side illuminated showing [-craters & textures,-] [+craters,+] a dark void separates it from ..."

**Spatial**

"... a dark void separates it from a lit spacecraft [-launching,-] [+launching slightly above,+] set against .."

**Contextual**

"... lit spacecraft launching, set against a starry cosmic [-background.-] [+background at twilight.+]"

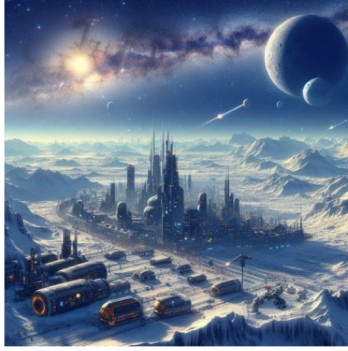

"In a futuristic setting, a lit city stands amidst a stark white landscape; above it swirls a galaxy of planets, moons, stars, and nebulas."

**Content**

"... above it swirls a galaxy of planets, moons, stars, and [+two+] nebulas."

**Attribute**

"In a futuristic setting, a lit city stands amidst a [-stark-] [+pristine+] white landscape;..."

**Spatial**

"... stark white landscape; [+around+] [-above-] it swirls a galaxy of planets, moons,..."

**Contextual**

"... stands amidst a stark white [-landscape-] [+landscape dotted with sparse vegetation+]; ..."

Figure 8: More Examples of prompt editing.

## A.5 MODIFICATION PROMPT INSTRUCTION FOR LLM

To generate the negative prompts $c^l$ for alignment training in our method, we have applied the Gemini 2.0 Flash model (Team et al., 2023) to modify the original prompts. Each modification is instructed to edit by using one or more editing strategies where each of these strategies following one of the modification principles described in Sec. 3.

We show how we instruct the Gemini 2.0 Flash model to modify the original prompt to generate $c^l$. The left side of Fig. 9 shows how we instruct Gemini model to modify a given prompt. The right-hand side shows the choices of modification strategy, which correspond to the four modification principles.

*Modification Instruction prompts*

You are an assistant tasked with modifying the description prompt of a image so that the meaning of the two prompts are different, but the images generated from the two prompts are hard to distinguish.When provided with a text-to-image generation prompt, please modify the prompt and change only one element of the prompt using the following strategy.The goal here is to generate a modified prompt that is similar to the original prompt, but conceptually different, to trick the model to generate the same image.
{strategy}

Please avoid color changes.
The image described by the modified text-to-image generation prompt and the original prompt should not be exactly the same, but visually similar.
Please output the modified text-to-image generation prompt, nothing else.
Here is the original text-to-image generation prompt:

{{prompt}}

Here is the modified text-to-image generation prompt:

*Modification Strategies*

Addition: Inserting new elements (objects, characters, landmarks) that weren't originally described.
Replacement/Substitution: Changing one element for another.
Quantitative Changes: Modifying the count or number of objects.
Appearance/Properties: Adjusting visual details such as texture, size, and material.
Style: Changing the artistic style or medium or adjusting descriptive adjectives to alter mood or tone.
Textual Detail: Enriching the description with additional adjectives or qualifiers to refine the prompt language and enhance realism or artistic expression.
Motion/Pose: Changing the dynamic state or pose of subjects.
Relations and Composition: Modifying spatial relationships among objects such as alignment, proximity, or composition.
Background: Altering or enriching the background elements.
Weather and Atmosphere: Introducing or modifying weather conditions.
Time and Lighting: Setting the scene at a specific time of day, season, or adjusting lighting conditions.
Setting: Changing the physical or cultural context

Figure 9: Left: the prompt template provide to the LLM to modify the a given prompt from image-prompt pair. Right: the precise way of modification that the LLM should apply for the given prompt.

## A.6 T2I ALIGNMENT'S STRONG CORRELATION WITH HUMAN PREFERENCE

In Sec. 4.3, we demonstrated that the implicit preference score is strongly negatively correlated with all human preference score metrics. Intuitively, the implicit preference score is defined as the difference in diffusion loss between positive and negative image–prompt pairs. It quantifies how much more likely the model is to generate image $x$ given its matching prompt $c^w$ compared to the mismatched prompt $c^l$. This loss therefore measures the model's ability to capture text–image alignment: models with higher implicit preference score also produce images that receive higher human preference scores. Consequently, **part of human preference can be attributed directly to better text–image alignment**.

To illustrate this, Figure 11 shows examples from the open-image-preferences-v1 dataset (Berenstein et al., 2024). Each pair compares two images generated from the same prompt: the left one contains the human-preferred image, and the right one the non-preferred image. Although all of these outputs are high-quality generations from state-of-the-art T2I models, annotators consistently favor the images that more faithfully reflect the prompt. For instance, in the first pair only the left image depicts the little girl "Alice." In the second pair the left image shows two portals, whereas the right shows only one. In the third pair only the left image appears to float in the sky. In the fourth pair the two androids hold hands in the left image but merely stand together in the right.

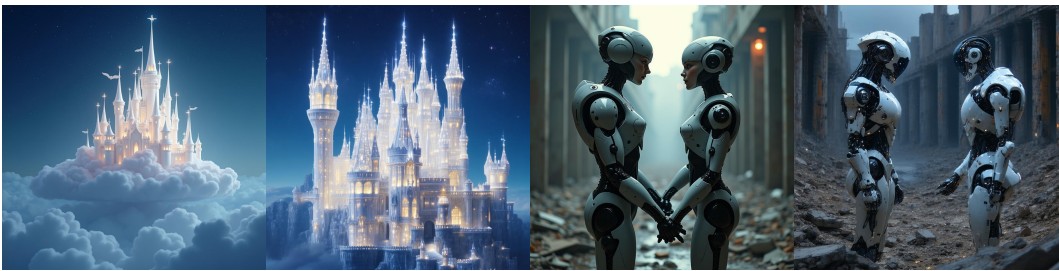

Figure 10: Left: Alice in a vibrant, dreamlike digital painting inside the Nemo Nautilus submarine. Right: An ethereal double portal with two paths illuminated by soft golden hour light, figures poised at the gateway in dynamic perspective, rendered in vibrant digital hues and sharp lines, with a serene mood and rich textures

Figure 11: Left: A glowing white fantasy castle with towers and spires, floating in a starlit sky. Right: Two androids hold hands, gazing into each other's eyes, in the ruins of a dystopian world, dramatic lighting.

These qualitative examples, together with our quantitative ablation study of implicit preference score versus human preference scores, suggest that human judgments are closely tied to text–image alignment.

### A.7 USER STUDY

We have conducted human study through user survey to evaluate the performance of our approach. An example of the user interface for our human study is shown in 12. During the study, we choose a subset of 100 prompts from PARTI-PROMPT dataset for evaluation. For each prompt, a pair of images generated by different methods from the prompt will be presented to the human subject. The human subject is then asked to select which image aligns better with the prompt, or if there is a tie. We gurantee that each pair of images are evaluated by at least 3 different human subjects.

### A.8 TRAINING DYNAMIC

**Training Curves of TKTO and TDPO** As shown in Fig. 13, we plot the training dynamics of the Implicit Preference Score defined in Sec. 4.3, together with the curves of the alignment metrics Pick Score, CLIP, and Human Preference Score on a held-out validation set. Note that the TDPO loss objective in Eq. (10) can be written as $\mathcal{L}_{\text{TDPO}}(\theta) = -\mathbb{E}\big[\log \sigma\big(\alpha\,\text{IPS}(\theta) + c\big)\big]$, where $\alpha = \beta, \omega(t) > 0$ and $c$ is a constant determined by the reference model, independent of $\theta$. Since the log-sigmoid function $\log \sigma(\cdot)$ is monotonically increasing and $\alpha > 0$, minimizing the TDPO loss is equivalent to maximizing the Implicit Preference Score; consequently, as the training loss decreases, the Implicit Preference Score increases.

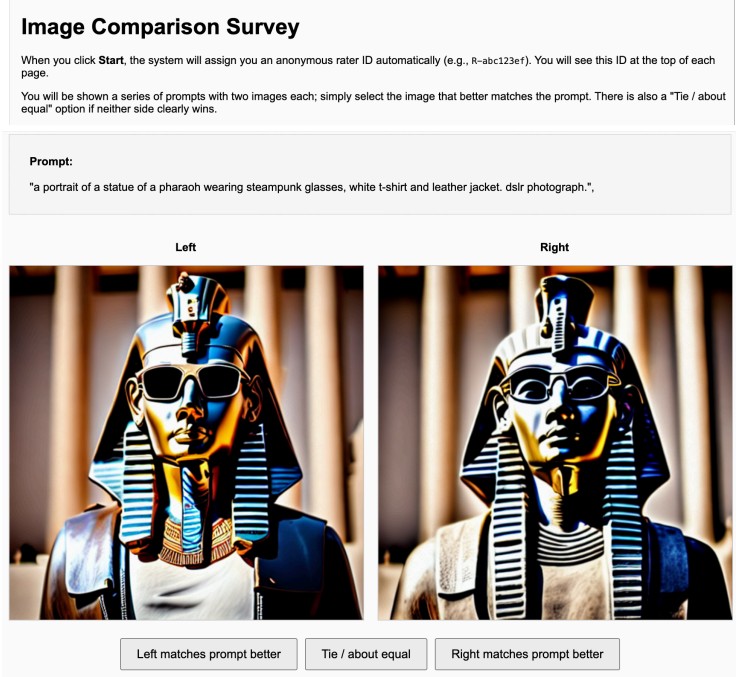

Figure 12: User study interface.

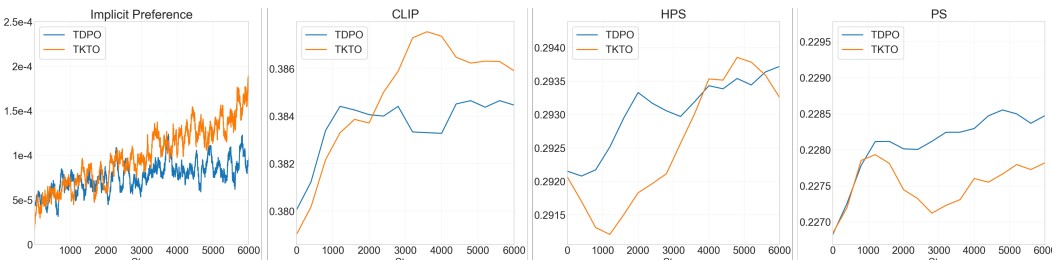

Figure 13: Training Curve

# B MORE EXPERIMENT RESULTS

## B.1 CLIPPING NEGATIVE GRADIENT

In our experiments, we observed that direct fine-tuning with objectives in the loss of TDPO and TKTO objectives would degrade the quality of the sample after several hundred steps. In our experiments, we found that directly optimizing towards the optimization objectives of TDPO and TKTO lead to unstable training. This is likely caused by the variance introduced by the negative gradient, a phenomenon extensively analyzed in prior work (Pal et al., 2024; Ren & Sutherland, 2025). To mitigate this , we introduce a clipping trick inspired by Proximal Policy Optimization (Schulman et al., 2017). Specifically, we clamp the squared L2 norm of the negative-sample term in the loss, which bounds extreme negative signals and stabilizes training. For example, in TDPO, we add a clamp function to the L2 squared norm on the $\theta$ parameter term condition on $\boldsymbol{c}^l$:

$$
\begin{aligned}
L_{\text{diff-tdpo}} = -\mathbb{E}_{(\boldsymbol{x}_0,\boldsymbol{c}^w,\boldsymbol{c}^l)\sim D,t\sim\mathcal{U}(0,T),\boldsymbol{x}_t\sim q(\boldsymbol{x}_t|\boldsymbol{x}_0)}[\log\sigma(\\
-\beta w(t)(||\boldsymbol{\epsilon}_\theta(\boldsymbol{x}_t,\boldsymbol{c}^w,t)-\boldsymbol{\epsilon}^w||_2^2 - ||\boldsymbol{\epsilon}_{\text{ref}}(\boldsymbol{x}_t,\boldsymbol{c}^w,t)-\boldsymbol{\epsilon}^w||_2^2\\
-(\text{clamp}(||\boldsymbol{\epsilon}_\theta(\boldsymbol{x}_t,\boldsymbol{c}^l,t)-\boldsymbol{\epsilon}^l||_2^2,\ \max=||\boldsymbol{\epsilon}_{\text{ref}}(\boldsymbol{x}_t,\boldsymbol{c}^l,t)-\boldsymbol{\epsilon}^l||_2^2+\lambda_{\text{bound}})\\
-||\boldsymbol{\epsilon}_{\text{ref}}(\boldsymbol{x}_t,\boldsymbol{c}^l,t)-\boldsymbol{\epsilon}^l||_2^2)))],
\end{aligned}
\tag{14}
$$

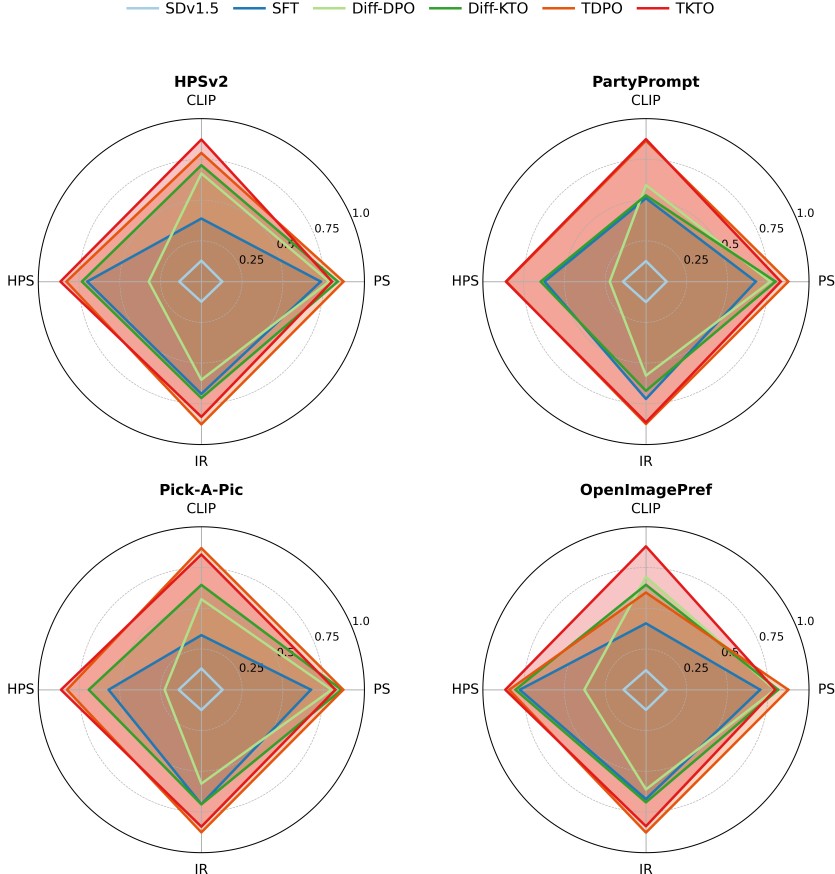

Figure 14: Radar chart of average scores of each model on each metric. The average scores are calculated by a relax min-max normalization and shifting the lower bound by one standard deviation to keep all curves in view. It shows that our methods (TDPO and TKTO) outperform the baselines (Diffusion-DPO, Diffusion-KTO, SFT and SD v1.5) on all metrics by covering all the baselines.

where $\lambda_{\text{bound}}$ is a hyper-parameter controls the strength of the maximum negative signal. For TKTO, this is applied similarly; the clamp function would be added when the condition prompt is not matched to the given image ( when $\omega(c) = -1$ ).

## B.2 HUMAN PREFERENCE METRIC PERFORMANCE

In addition to win rates, we also report the average scores of each model on each metric in a radar chart Fig. 14 for each dataset. We can see that our methods cover almost all the baselines on all metric dimensions and all dataset, suggesting that our methods have a higher average score for almost all the metric in all evaluation dataset.

## B.3 EXPERIMENT ON SDXL

Additional SDXL results are presented in Tab. 4. We adopt the same evaluation configuration as Tab. 2: competing methods are trained using both preferred and dispreferred images from Pick-a-Pic, whereas our method relies solely on the winning image. The results indicate that our method achieves superior Image Reward and CLIP scores, while performing less strongly on HPS and Pick Score. Importantly, it provides a clear improvement over the SDXL base model and surpasses the SFT baseline, whose performance deteriorates due to the lower visual quality of Pick-a-Pic images compared with SDXL's base outputs.

Table 4: Alignment Performance comparison of SDXL. Win rates (%) are reported against SDXL.

| Method | Supervision | PICK-A-PIC | | | | PARTI-PROMPT | | | |
|---|---|---|---|---|---|---|---|---|---|
| | | PS | CLIP | HPS | IR | PS | CLIP | HPS | IR |
| SFT* | $(x, c)$ | 20.80 | 44.80 | 40.60 | 34.40 | 17.03 | 36.58 | 33.02 | 37.18 |
| Diff.-DPO* | $(x^w, x^l, c)$ | **75.20** | 59.40 | 76.20 | 65.20 | 65.44 | 60.54 | 74.08 | 66.85 |
| MaPO* | $(x^w, x^l, c)$ | 54.40 | 51.20 | 69.60 | 61.40 | 58.34 | 47.43 | 66.54 | 58.64 |
| DSPO* | $(x^w, x^l, c)$ | 74.00 | 59.60 | **80.00** | **68.60** | **67.46** | 55.02 | **81.80** | **73.47** |
| Ours-TDPO-SDXL | $(x, c^w, c^l)$ | 42.60 | **62.60** | 58.20 | 65.60 | 47.25 | **63.38** | 64.94 | 67.56 |
| Ours-TKTO-SDXL | $(x, c^w, c^l)$ | 56.40 | 61.00 | 70.40 | 67.00 | 57.19 | 58.31 | 68.25 | 68.38 |

Table 5: Ablation on rewriting LLM's effect on the final performance.

| Method | Rewriting LLM | PICK-A-PIC | | | | PARTI-PROMPT | | | |
|---|---|---|---|---|---|---|---|---|---|
| | | PS | CLIP | HPS | IR | PS | CLIP | HPS | IR |
| SFT | – | 71.20 | 50.80 | 62.00 | 72.80 | 69.06 | 52.38 | 63.31 | 73.81 |
| TDPO | Gemini-Flash-2.0 | 75.20 | 56.00 | 69.20 | 82.60 | 74.63 | 57.25 | 70.19 | 78.75 |
| | Qwen3-30b-a3b-it | 75.20 | 54.80 | 64.40 | 78.40 | 72.19 | 54.94 | 65.19 | 76.25 |
| | Llama-3.1-8b-it | 68.20 | 54.00 | 61.00 | 72.80 | 67.44 | 54.00 | 64.44 | 75.81 |

### B.4 MORE ABLATION STUDY: THE EFFECT OF LARGE LANGUAGE MODEL CHOICE FOR REWRITING

We examine the effect of choosing a large language model for prompt rewriting. Specifically, we choose to use open-source alternatives Qwen3-30B-A3B-it and Llama-3.1-8b-it models. While Qwen3-30B-A3B-it is generally more capable and has a close LLM Arena score(LMA, 2023) as compared to Gemini-Flash-2.0, Llama-3.1-8b-it is less capable. We show the performance comparison of using these three models for generating negative prompts as follows. It can be seen that while Gemini-Flash-2.0 produce the best performance . Llama-3.1-8b-it fails to provide meaningful improvements over the supervised-finetuning baseline. This suggests that the general capability of the LLM has a non-negligible impact on the final alignment performance. Notably, open-source alternatives such as Qwen3-30b-a3b-instruct are sufficiently powerful to deliver improvements over the baseline methods.

### B.5 MORE QUALITATIVE COMPARISON

In this subsection, we put more qualitative comparison images in Fig. 15

### B.6 MORE ABLATION STUDY: MODIFICATION BUDGET

For each experiment, we first choose an editing budget $k \in 1, 2, 3$. Then, for each $c$, we perform k edits: at each step we randomly select one of the four modification principles and prompt the language model to apply its corresponding editing strategy to the prompt.

For both TDPO (Eq. (10)) and TKTO (Eq. (11)), we vary the editing budget $k \in \{1, 2, 3\}$—applying $k$ distinct prompt-editing strategies to generate negatives—and denote variants as c1, c2, c3.

Here we raise the question: what if we have multiple changes in the text prompt, that is, what is the effect of increasing the difference between the positive prompt $x^w$ and negative prompt $x^l$. Our ablation study of the budget of prompt-editing strategies shows that our methods behave differently when we have different modification budgets. The plot in Fig. 16 reports the normalized metric scores for each of the PickScore, CLIP, HPS and ImageReward. The plot shows neither consistent improvement nor consistent degrading performance. Interestingly, certain metrics seem to benefit from having larger modification budgets; this is an interesting observation and we leave this for future work.

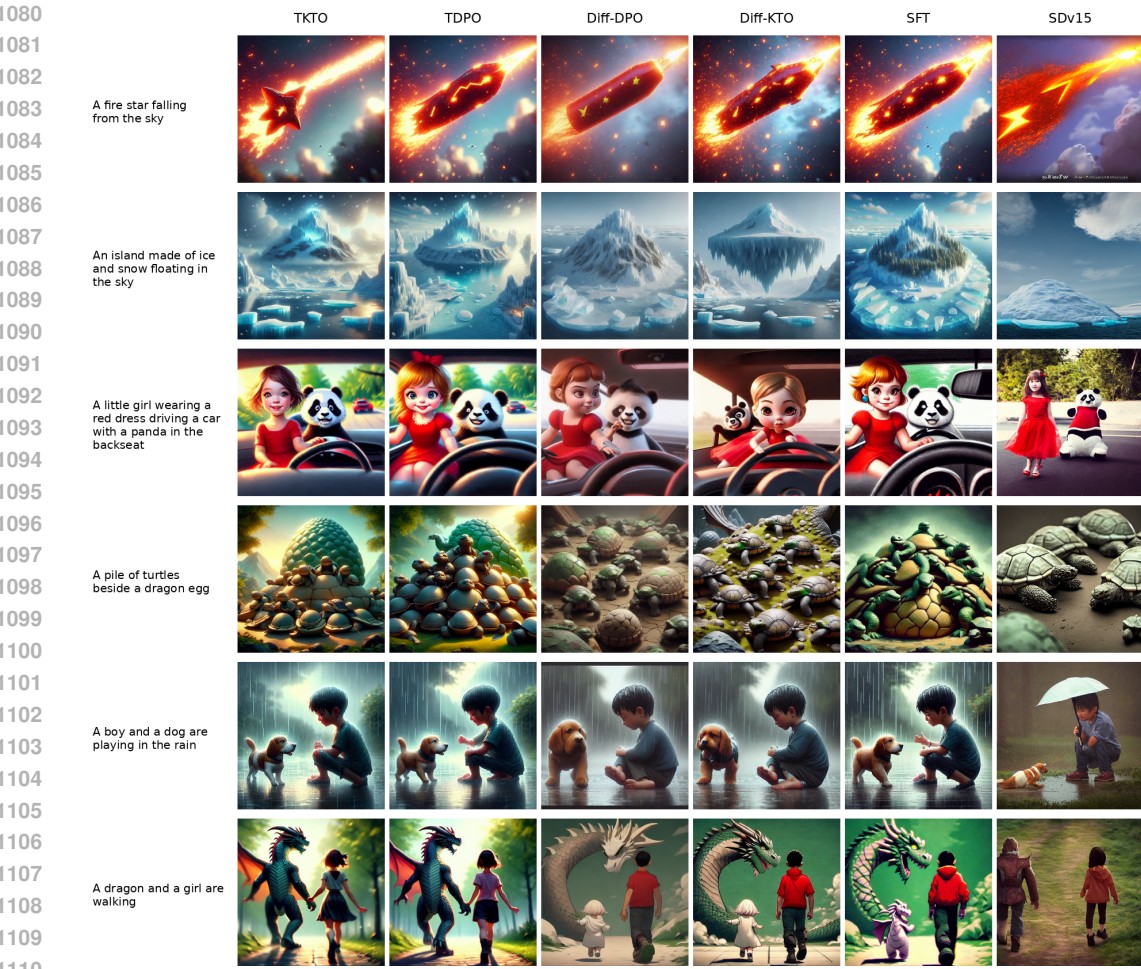

Figure 15: Side-by-Side grid comparison of the image generation using our methods and the baselines. The left most column show the prompts used to generate the images.

### B.7    DIFF-DPO AND DIFF-KTO BASELINES

The images in our training dataset HPSD (Egan et al., 2024) are of exceptionally high quality. However, they can't be directly used by many preference alignment method. Methods such as Diffusion-DPO (Wallace et al., 2024) and Diffusion-KTO (Li et al., 2024) require human preference labels: they assume that each image pair is annotated with a preferred example. In contrast, HPSD provides neither paired images nor preference annotations. To be able to compare our method with these baselines, we need to generate the second image in this dataset. We build our experiments according to (a), (c), (d) cases in Fig. 4. These cases summarize how we could finetune our model when **a human preference annotation or image pair is absent**.

- (a) SFT: This is the straightforward baseline where we finetune our model on the HPSD dataset using the regular diffusion loss. We denote this baseline as **SFT** in Table 6

- (c) Constructed Image Pair: This case cover how we are going to use Diffusion-DPO and Diffusion-KTO in this setting. We conduct experiment under three different setting of how we generate the "negative" sample for the image pair. **Diff-DPO-sd 35/Diff-KTO-sd35**: refer to the case where we first construct a negative prompt according to the four modification principles. Then we use the production level open-source T2I model, Stable Diffusion v3.5m to sample a image according to that prompt. This sampled image generally has a relatively high quality. **Diff-DPO-SFTsd15/Diff-KTO-SFTsd15**: In this case, we also modified the prompt first and sample an image base on this modified prompt, except this time we sampled by feeding the Stage 1 SDv1.5

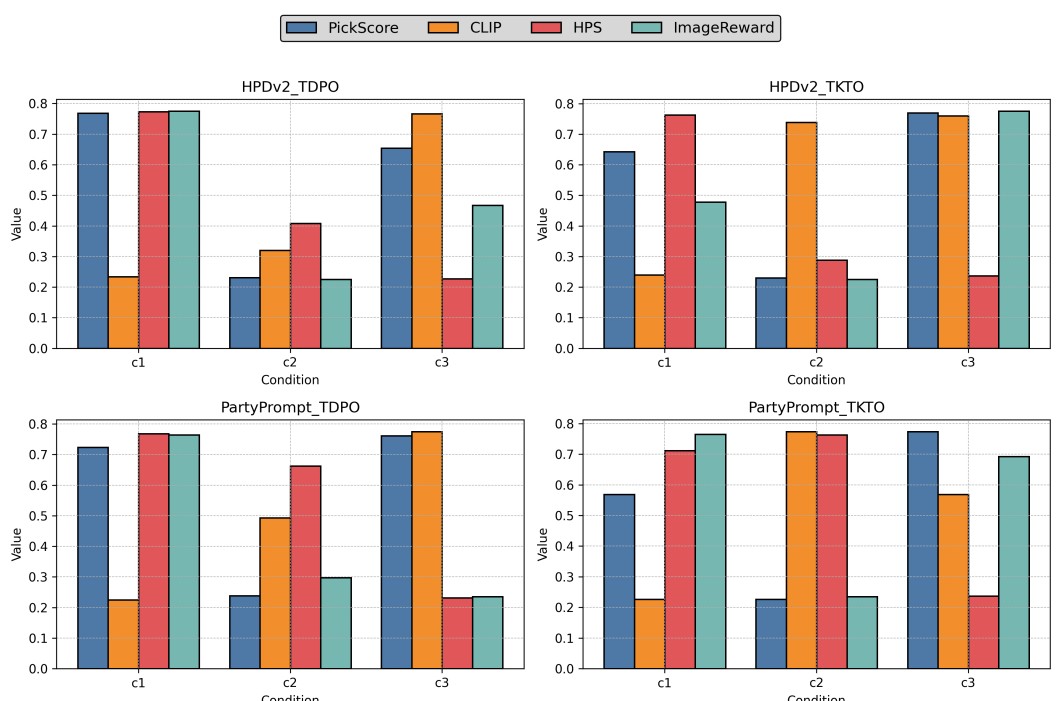

Figure 16: Normalized Metrics given different modification budgets in HPDv2 and PartiPrompts.

finetuned model checkpoint with the same HPSD prompts. This sampled image generally has lower quality than SD3.5, but should have a closer distribution to the training set HPSD as the model is finetuned on it, which may align better to the default setup of Diffion-DPO and Diffuion-KTO. **Diff-DPO-quality/Diff-KTO-quality**: In this case, we don't construct a modified prompt, we directly use the original prompt to construct our "negative" sample using the SFT sd15 model. This sample is "negative" because of the fact that they are less visually appealing than the original, high-quality human-curated HPSD dataset, which were produced by a stronger base model (DALL·E 3) (OpenAI, 2023). This makes them naturally fall into the category of not preferred images.

- (d): Constructed Text Pair: In this setting, we don't generated the "negative" image sample, we only construct a modified prompt, and using TDPO and TKTO objective function to finetune our model. We denote this method as **Ours-TDPO/Ours-TKTO**.

We put all the result in Tab. 6. It shows that our method, TDPO and TKTO, consistently outperform most of these baselines, showing that our methods is a more promising approach when an image pair or preference label is absent.

## C  THE USE OF LARGE LANGUAGE MODELS (LLMs)

In this work, LLMs primarily serve as a prompt-editing tool. Details are provided in Appendix A.5. We also used them to improve writing by refining the flow of paragraphs and correcting grammar errors. It serves as an additional tool for polishing paper writing, but is not the main writer of this work.

## D  LIMITATION

Our preference-data-free alignment framework, while effective, has several limitations. First, its success hinges on the quality of prompt editing: our current strategies may overlook subtle semantic distinctions or produce unnatural negative examples. Second, because we rely on a budget-constrained off-the-shelf LLM, generated negatives can suffer from reduced fluency and faithfulness, which

Table 6: Comparison of Win Rates Across Datasets and Methods. The best results are highlighted in **bold**, and the second-best results are underlined. Baseline methods are shown in normal font, while our methods are highlighted in blue. It shows that our methods consistently outperform the baselines across all datasets for most metrics except for a few entries.

| Dataset | Method | PS | AES | CLIP | HPS | IR |
|---|---|---|---|---|---|---|
| HPSv2 | SFT | 76.25 | 66.75 | 52.00 | 75.75 | 76.00 |
| | Diff-DPO-sd35 | 79.5 | 65.75 | 49.00 | 76.5 | 74.5 |
| | Diff-KTO-sd35 | 77.25 | 69.50 | 52.25 | 72.25 | 75.00 |
| | Diff-DPO-SFTsd15 | 77 | 70.75 | 54.75 | 59.00 | 70.00 |
| | Diff-KTO-SFTsd15 | 80.25 | 72.50 | 53.75 | 76.00 | 76.75 |
| | Diff-DPO-quality | 75.75 | 69.00 | **56.00** | 54.75 | 68.00 |
| | Diff-KTO-quality | 80.75 | 73.25 | 52.75 | 76.50 | 75.50 |
| | Ours-TDPO | **83.25** | **73.75** | **56.00** | 79.00 | **82.25** |
| | Ours-TKTO | 80.00 | 68.75 | 55.75 | **81.00** | 80.75 |
| PartyPrompt | SFT | 69.06 | 73.50 | 52.38 | 63.31 | 73.81 |
| | Diff-DPO-sd35 | 71.94 | 72.25 | 48.69 | 64.06 | 73.37 |
| | Diff-KTO-sd35 | 68.06 | 74.06 | 50.94 | 63.69 | 70.81 |
| | Diff-DPO-SFTsd15 | 71.06 | 72.31 | 54.06 | 49.94 | 64.88 |
| | Diff-KTO-SFTsd15 | 73.50 | 74.94 | 52.63 | 63.69 | 71.62 |
| | Diff-DPO-quality | 69.88 | 71.19 | 52.25 | 46.44 | 63.25 |
| | Diff-KTO-quality | 73.31 | 75.19 | 52.19 | 64.13 | 70.56 |
| | Ours-TDPO | **74.63** | 79.00 | 57.25 | 70.19 | **78.75** |
| | Ours-TKTO | 73.62 | **79.25** | **57.31** | **70.63** | 77.31 |
| Pick-A-Pic | SFT | 71.20 | 67.40 | 50.80 | 62.00 | 72.80 |
| | Diff-DPO-sd35 | 72.40 | 66.20 | 52.2 | 66.00 | 76.80 |
| | Diff-KTO-sd35 | 72.20 | 68.80 | 54.00 | 64.4 | 74.60 |
| | Diff-DPO-SFTsd15 | 72.80 | 70.00 | 54.20 | 52.40 | 68.60 |
| | Diff-KTO-SFTsd15 | **77.40** | 73.00 | 55.20 | 65.60 | 74.60 |
| | Diff-DPO-quality | 72.20 | 69.80 | 53.80 | 50.40 | 68.20 |
| | Diff-KTO-quality | 77.20 | **75.20** | 56.20 | 65.00 | 74.60 |
| | Ours-TDPO | 75.20 | 72.6 | 56.00 | 69.20 | **82.60** |
| | Ours-TKTO | 74.40 | 74.20 | **58.20** | **70.80** | 80.20 |
| OpenImagePref | SFT | 80.60 | 58.80 | 53.60 | 77.00 | 77.60 |
| | Diff-DPO-sd35 | 81.40 | 58.80 | 46.60 | 73.60 | 77.40 |
| | Diff-KTO-sd35 | 77.80 | 62.40 | 54.60 | 75.80 | 77.00 |
| | Diff-DPO-SFTsd15 | 83.60 | 69.80 | 57.20 | 62.00 | 73.20 |
| | Diff-KTO-SFTsd15 | 82.60 | 66.60 | 57.00 | 76.60 | 80.80 |
| | Diff-DPO-quality | 83.60 | 68.00 | 56.40 | 58.80 | 73.80 |
| | Diff-KTO-quality | 82.60 | 68.20 | 57.40 | 78.60 | 78.60 |
| | Ours-TDPO | **88.20** | **70.60** | 58.20 | **80.20** | **86.00** |
| | Ours-TKTO | 84.20 | 68.40 | **60.80** | 79.20 | 82.40 |

in turn can degrade alignment performance. Third, we fine-tune only the diffusion model while keeping the pre-trained text encoder fixed; this static encoder may limit the framework's capacity to discriminate between closely related prompts. Finally, generating all negatives from a single LLM restricts the diversity of hard negatives—incorporating adversarial, retrieval-based, or multi-model sampling strategies could further improve robustness.

# E    REPRODUCIBILITY STATEMENT

Detail setup of our method, model architecture, training and evaluation pipeline has been outlined in the work. We also have a consistent training and evaluation setup for better reproduction. Moreover, the code for this work will be release and available soon.

