# OpenReview forum: "Free Lunch Alignment of Text-to-Image Diffusion Models without Preference Image Pairs"
_ICLR.cc/2026/Conference — Submitted to ICLR 2026_

### Official Review · Reviewer_6UbU · 2025-10-31

**Soundness:** 3
**Presentation:** 3
**Contribution:** 3
**Rating:** 6
**Confidence:** 3

**Summary:**

This paper proposes Text Preference Optimization (TPO) for aligning text-to-image (T2I) diffusion models without human preference image pairs. It creates matched/mismatched text pairs via LLM-based prompt editing under four principles (content, attribute, spatial, contextual). Two variants, TDPO and TKTO, adapt DPO/KTO to diffusion training. Experiments across several benchmarks indicate consistent gains over Diffusion-DPO/KTO while remaining annotation-free.

**Strengths:**

1. The idea of "free-lunch" alignment (re-using existing caption data + LLM perturbations) is creative, practical, and addresses real scalability issues in diffusion-model alignment.
2. The formulation of TDPO/TKTO directly mirrors established DPO/KTO objectives, but shifts preference comparison from outputs (images) to inputs (text prompts).
3. The method has been evaluated on four diverse datasets with multiple quantitative metrics. It achieves consistent improvement across most metrics and datasets. Qualitative comparisons convincingly show better text-image faithfulness, and ablation studies on editing principles and implicit-preference correlation provide solid analytical depth.

**Weaknesses:**

1. The pipeline is essentially "synthetic preference data + existing DPO/KTO objective." Although effective and creative, it feels more like an engineering simplification than a fundamentally new learning principle.
2. The framework depends on a single budget-constrained LLM for prompt editing, which may introduce stylistic or semantic biases and restrict negative-sample diversity.

**Questions:**

1. Can authors please clarify how LLM-generated and human-labeled prompts compare in alignment performance, possibly via a short correlation or qualitative analysis. This would clarify whether TPO truly approximates human alignment behavior or just benefits from larger-scale synthetic coverage.
2. I have concerns with the dependence on LLM prompt quality. Alignment quality is tied to how well the Gemini 2.0 Flash edits produce meaningful yet challenging negatives. Are there any quantitative analysis of negative-sample diversity?

---

> ### Author Response · Authors · 2025-11-21
> **(2/3) Response to Reviewer 6UbU**
>
> > W2 \& Q2: The framework depends on a single budget-constrained LLM for prompt editing, which may introduce stylistic or semantic biases and restrict negative-sample diversity. I have concerns with the dependence on LLM prompt quality. Alignment quality is tied to how well the Gemini 2.0 Flash edits produce meaningful yet challenging negatives. Are there any quantitative analyses of negative-sample diversity?
>
> **R2:**
> We thank the reviewer for raising this interesting and important question.
> In practice, we find that prompt editing is a relatively easy task, where a budgeted LLM already performs well at producing fine-grained and high-quality prompt modifications. Our quantitative results in table 1 and 2 suggest that the proposed method works effectively under such a budgeted LLM. At the same time, we agree with the reviewer that the capability and style of the LLM can influence the quality/diversity of the synthetic dataset and, consequently, the alignment behavior of TPO.
>
> Taking this concern seriously, we plan to empirically examine the impact of LLM choice. Specifically, we will include an ablation using two additional LLMs of different capacities, qwen3-30b-a3b-instruct-2507 and llama-3.1-8b-it. LLM arena indicates a general capability of llama-3.1-8b-it < qwen3-30b-a3b-instruct-2507 $\approx$ gemini-2.0-flash. For each model, we (1) regenerate the modified prompts (2) compare the diversity within the modified prompts (3) rerun finetuning.
>
> ~~Due to limited computational resources, the experiment is still ongoing.~~
>
> **R2.1** To answer your first question about LLM's capabilities' effect on final performance,
> We show the performance comparison of using these three models for generating negative prompts as follows.
>
> **Dataset: Pick-a-Pic**
>
> | Method | Rewriting LLM         | PS    | CLIP  | HPS   | Image Reward |
> |--------|------------------------|-------|-------|-------|--------------|
> | SFT    | –                      | 71.20 | 50.80 | 62.00 | 72.80        |
> | DiffusionDPO| -  | 72.80 | 54.20 | 52.40 | 68.60 |
> | TDPO   | Gemini-Flash-2.0       | 75.20 | 56.00 | 69.20 | 82.60        |
> | TDPO   | Qwen3-30b-a3b-it       | 75.20 | 54.80 | 64.40 | 78.40        |
> | TDPO   | Llama-3.1-8b-it        | 68.20 | 54.00 | 61.00 | 72.80        |
>
>
> **Dataset: Parti-Prompt**
>
> | Method | Rewriting LLM         | PS    | CLIP  | HPS   | Image Reward |
> |--------|------------------------|-------|-------|-------|--------------|
> | SFT    | –                      | 69.06 | 52.38 | 63.31 | 73.81        |
> |DiffusionDPO| - | 71.06 | 54.06 | 49.94 | 64.88|
> | TDPO   | Gemini-Flash-2.0       | 74.63 | 57.25 | 70.19 | 78.75        |
> | TDPO   | Qwen3-30b-a3b-it       | 72.19 | 54.94 | 65.19 | 76.25        |
> | TDPO   | Llama-3.1-8b-it        | 67.44 | 54.00 | 64.44 | 75.81        |
>
> It can be seen that Gemini-Flash-2.0 produces the best performance followed by Qwen3-30b-a3b-it. Llama-3.1-8b-it fails to provide meaningful improvements over the supervised-finetuning baseline. This suggests that the general capability of the LLM has a non-negligible impact on the final alignment performance. Notably, open-source alternatives such as Qwen3-30b-a3b-instruct are sufficiently powerful to deliver improvements over the baseline methods.
>
>
> **R2.2** To further investigate the cause and answer your second question about negative sample quality, we furhter investigate the difference between the negative samples generated by different rewritting llms.
>
> | Metrics | Gemini-Flash  | Qwen3 | Llama |
> |-|-|-|-|
> | Performance(Average) | 70.5 | 68.3 | 64.7|
> | Prompt Difference(Edit Distance)| 2.8 | 8.4 | 19.3 |
> | Diversity(Edit Distance)| 4.9 | 13.2 | 20.1 |
> | Base model Implicit Preference Score $(10^{-5})$ | 5.2  | 6.5  | 12.6  |
>
> As shown in the table above, prompt difference measures the average edit distance between the negative prompt and the original prompt, while negative diversity is computed as the mean pairwise edit distance among negative prompts within each group. Interestingly, we observe a negative correlation between performance and prompt difference—models that produce smaller edits to the original prompt tend to achieve higher performance. Since this trend is not intuitive, we further examine the corresponding Implicit Preference Scores of the prompt pairs on the base model.
>
> **We find that better-performing LLMs with lower edit distances consistently exhibit lower Implicit Preference Scores on T2I base model(sd 1.5). This means the T2I base model has a harder time telling the prompts apart, suggesting that these LLMs generate more nuanced, fine-grained modifications and that this subtlety is associated with improved downstream performance.**
>
> We've add new result to the revised paper.
> [1] https://lmarena.ai/leaderboard

---

> ### Author Response · Authors · 2025-11-21
> **(3/3) Response to Reviewer 6UbU**
>
> > Q1: Can authors please clarify how LLM-generated and human-labeled prompts compare in alignment performance, possibly via a short correlation or qualitative analysis. This would clarify whether TPO truly approximates human alignment behavior or just benefits from larger-scale synthetic coverage.
>
> **R3:**
> We interpret the question as asking whether human annotators could produce negative prompts, and how our synthetically generated negatives compare to such human-modified negative prompts.
> Firstly, We would like to clarify that, in our work, **“human alignment behavior” refers to human preferences over images given a single prompt**. DiffusionDPO and DiffusionKTO require human-annotated image pairs conditioned on the same prompt, whereas our method utilizes synthetically generated text prompts. As shown in Table 2 and Figures 7 and 8, our Implicit Preference Score (see Section 4.3), is positively correlated with established human preference metrics (HPS, PS) trained from human-annotated image pairs. **This shows that our TPO aligns the model better to human preference.**
>
> Secondly, At the same time, there are several issues for training with human-labled prompts:
> * It is practically challenging to collect a sufficiently large and diverse set of human-written negative prompts.
> * When humans are instructed to create negative prompts, they tend to be less creative and often produce only trivial modifications. Training on such low-quality modifications does not necessarily lead to performance improvement, especially since the original model is already strong enough to distinguish these simple changes.
>
> Here we show a qualitative example where we asked human subjects to modify a prompt following our editing principles:
>
> ```
> Original_prompt = Waterfall cascading down a cliff forms an elephant's likeness in verdant surroundings, flowing into a river bordered by large stones beneath the filtered sunlight
> Human 1 Modification: elephant -> dog
> Human 2 Modification: river -> sea
> Human 3 Modification: large stones -> small stones
> LLM Modification 1: verdant surroudning -> lush surrounding
> LLM Modification 2: bordered by large stones ->  bordered by pebbles
> LLM Modification 3: beneath the filtered sunlight -> beneath soft morning light.
> ```
> We can see that LLM-based prompt edits tend to produce finer-grained, higher-quality modifications, whereas human edits are often limited to simple object or attribute substitutions.

---

> ### Author Response · Authors · 2025-11-29
> **(1/3) Response to Reviewer 6UbU**
>
> Weaknesses:
> > W1: The pipeline is essentially "synthetic preference data + existing DPO/KTO objective." Although effective and creative, it feels more like an engineering simplification than a fundamentally new learning principle.
>
> **R1:** We acknowledge the reviewer’s concern and appreciate the perspective. Traditionally, aligning a generative model to human preferences requires the costly and time-consuming collection of human preference data. In contrast, our method provides a new alignment pathway that does not rely on any human preference supervision. While our approach indeed combines synthetic preference data with existing DPO/KTO objectives, we respectfully argue that this is more than an engineering simplification: it introduces a new paradigm for text-to-image alignment. By leveraging any high-quality image–caption dataset to construct synthetic text pairs, our method can, in principle, be scaled up much more efficiently. We see this scalability as a key advantage and an exciting direction for future work.

---

### Official Review · Reviewer_fwFA · 2025-11-01

**Soundness:** 3
**Presentation:** 3
**Contribution:** 3
**Rating:** 6
**Confidence:** 5

**Summary:**

The authors propose TPO, a novel T2I alignment framework that perturbs the prompt with LLM to form a prompt level preference pair and aligns the Diffusion T2I model via prompt level preference. TPO achieved superior performance across multiple automatic feedback benchmarks on SD1.5 when integrated to DPO and KTO. In addition, TPO disentangles the need for image preference data in T2I alignment, further enhancing alignment efficiency.

**Strengths:**

* The TPO algorithm is a novel contribution for Diffusion T2I alignment leveraging prompt level preference rather than image level.
* The TPO algorithm achieved superior performance with less requirements on training data.

**Weaknesses:**

* My major concern is the robustness of TPO. The experiments are conducted only on SD1.5, one of the small scale open source T2I diffusion models. Comparison with DPO on SDXL are expected to demonstrate the robustness of this interesting algorithm.

* Evaluation from human or large VLM / MLLM is needed. The automatic evaluation metrics used in Table 1 and Table 2 are CLIP and fine-tuned CLIP/BLIP variants while SD1.5 uses CLIP as text encoder. The training recipe of TPO could potentially boost the similarity of the generations and the CLIP embedding and hence significantly improves performance on these related metrics. It would be good to see some evaluation that relies less on CLIP.

* In Table 1, there's a typo "PartyPrompt", please revise for "Parti-Prompt"

**Questions:**

See Weaknesses

---

> ### Author Response · Authors · 2025-11-21
> **Response to Reviewer fwFA**
>
> > W1. My major concern is the robustness of TPO. The experiments are conducted only on SD1.5, one of the small scale open source T2I diffusion models. Comparison with DPO on SDXL are expected to demonstrate the robustness of this interesting algorithm.
> A1.
>
> **R1:** Thank you for your suggestion. We have left a placeholder in Section 4.2 and plan to update the paper with additional results on SDXL as soon as possible. ~~Due to limited computational resources and the higher training cost of SDXL, the experiment is still ongoing.~~
> Additional SDXL results are presented in **Table 4** in the updated draft. We did not do hyper-parameter tuning and adopt the same training setting as SD-1.5 experiments due to the limited time. We adopt the same evaluation configuration as Table 2: competing methods are trained using both preferred and dispreferred images from Pick-a-Pic, whereas our method relies solely on the winning image. We briefly summarize the performance comparison as follows:
>
> **Pick-a-Pic — (Win rates against SDXL)**
>
> | **Method**            | **PS** | **CLIP** | **HPS** | **IR** |
> |-----------------------|--------|----------|---------|--------|
> | **SFT**               | 20.80  | 44.80    | 40.60   | 34.40 |
> | **Diff.-DPO**         | **75.20**  | 59.40    | **76.20**   | 65.20 |
> | **MaPO**              | 54.40  | 51.20    | 69.60   | 61.40 |
> | **Ours-TDPO-SDXL**    | 42.60  | **62.60**    | 58.20   | 65.60 |
> | **Ours-TKTO-SDXL**    | 56.40  | 61.00    | 70.40   | **67.00** |
>
>
> **Parti-Prompt — (Win rates against SDXL)**
>
> | **Method**            | **PS** | **CLIP** | **HPS** | **IR** |
> |-----------------------|--------|----------|---------|--------|
> | **SFT**               | 17.03  | 36.58    | 33.02   | 37.18 |
> | **Diff.-DPO**         | **65.44**  | 60.54    | **74.08**   | 66.85 |
> | **MaPO**              | 58.34  | 47.43    | 66.54   | 58.64 |
> | **Ours-TDPO-SDXL**    | 47.25  | **63.38**    | 64.94   | 67.56 |
> | **Ours-TKTO-SDXL**    | 57.19  | 58.31    | 68.25   | **68.38** |
>
>
> This results shows that our method achieves superior Image Reward and CLIP scores, while performing less strongly on HPS and Pick Score. Importantly, it provides an improvement over the SDXL base model (except for pick-score on TDPO) and surpasses the SFT baseline, whose performance deteriorates due to the lower visual quality of Pick-a-Pic images compared with SDXL’s base outputs.
>
> > W2. Evaluation from human or large VLM / MLLM is needed. The automatic evaluation metrics used in Table 1 and Table 2 are CLIP and fine-tuned CLIP/BLIP variants while SD1.5 uses CLIP as text encoder. The training recipe of TPO could potentially boost the similarity of the generations and the CLIP embedding and hence significantly improves performance on these related metrics. It would be good to see some evaluation that relies less on CLIP.
>
> **R2:** Thanks for valuable insignts. **We conduct a human study comparing DiffusionDPO, supervised fine-tuning (SFT), and our TDPO method, focusing on prompt–image alignment, and have updated the corresponding result in Sec 4.2 of the paper.** For DiffusionDPO, we use official checkpoint trained on the Pick-a-Pic dataset. For TDPO, we adopt the Table 2 setting, where only the winning images from Pick-a-Pic are used. For the SFT baseline, we fine-tune SD1.5 on the Pick-a-Pic winning images. This setup ensures a fair comparison, as all methods are trained on the same set of winning image–prompt pairs.While DiffusionDPO additionally has access to human-dispreferred images, TDPO instead leverages generated negative prompts.
>
> For human studeis we choose a subset of 100 prompts from PARTI-PROMPT dataset. During the study,a pair of images generated by different methods from the sample prompts will be presented, the human subject was asked to select which image aligns better with the prompt or if there is a tie. We gurantee that each pair of images are evaluated by at least 3 different human subjects. From Figure 6,It can been seen that our TDPO is favored over DiffusionDPO and SFT baselines with a win rate of **65.5%** and **67.62** respectively.
>
>
>
> > W3: In Table 1, there's a typo "PartyPrompt", please revise for "Parti-Prompt"
>
> **R3:** Thanks for spoting this, we have correct this in the revised paper.

---

### Official Review · Reviewer_PZtZ · 2025-11-01

**Soundness:** 2
**Presentation:** 3
**Contribution:** 3
**Rating:** 4
**Confidence:** 4

**Summary:**

This paper introduces Text Preference Optimization (TPO), a framework to align text-to-image diffusion models without requiring human-annotated image preference pairs. The method uses a Large Language Model (LLM) to generate mismatched text prompts from original captions, creating text preference pairs (matched vs. perturbed) to train the model. The authors extend DPO and KTO to TDPO and TKTO, claiming state-of-the-art results on benchmarks like HPSv2 and Pick-a-Pic. While the core idea of "free lunch" alignment is novel, the paper's claims are significantly undermined by critical flaws in the experimental setup and evaluation.

**Strengths:**

1. Original Idea: The concept of aligning models via automatically constructed text preferences, rather than image preferences, is novel and creative.
2. Practical Motivation: The work tackles a key scalability bottleneck in T2I alignment, offering a potentially cost-effective solution.
3. General Framework: The approach is model-agnostic and can be integrated with various preference optimization algorithms (DPO, KTO), which is a strong design choice.

**Weaknesses:**

- Unconvincing Baseline Performance: The dramatic drop in performance for the Diffusion-DPO baseline (e.g., HPS score) is a major red flag. It suggests either a flawed implementation, inappropriate hyperparameters, or an unfair comparison setup, which invalidates the claimed superiority of the proposed methods.
- Opaque Training Process: The paper provides no insight into the training dynamics. There are no loss curves, convergence plots, or monitoring metrics on held-out benchmarks. This makes it impossible to assess if the models are learning the intended preferences or if the training is stable.
- Potentially Unfair Comparison: The baselines (Diffusion-DPO/KTO) are designed for human image preferences. Comparing them against a method using synthetic text preferences, where the negatives are generated by the same LLM pipeline, may not be a fair or meaningful comparison. The paper should have included a baseline using the same synthetic text to generate a negative image for a more controlled comparison.

**Questions:**

See weakness.

---

> ### Author Response · Authors · 2025-11-21
> **(1/2) Response to Reviewer PZtZ**
>
> We appreciate your comments and concerns of our work. Before we address your concerns, we would like to start with a clarification on baseline settings.
>
> **Clarifications**:
>
> **C1:  Our DiffusionDPO/KTO Baselines reported in Table 1 use a fair sythetic-image setting**
> In Table 1. To fairly compare with our method that train on our dataset, we train DiffusionDPO and DiffusionKTO by constructing paired preference images from our source data. (As shown in Figure 4.c) Using the original image from the human-curated HPSD image caption dataset as the wining image, we tried 3 diffrent variants of constructing image pairs for DiffusionDPO and DiffusionKTO
>
>  1.	*Diff-DPO-sd35 / Diff-KTO-sd35*: Using the negative prompt generated by our method, we employ Stable Diffusion v3.5 to generate the corresponding losing image.
>
>  2.	*Diff-DPO-SFTsd15 / Diff-KTO-SFTsd15*: Similar to (1), we use the negative prompt, but generate the losing image with an SD v1.5 model finetuned on HPSD.
>
>  3.	*Diff-DPO-quality / Diff-KTO-quality*: Assuming human-curated HPSD images are superior to those from the SFT sd15 model, we directly use images generated by SFT sd15 from the original prompts (no modification) as lower-quality “negative” samples.
>
> **C2: Our DiffusionDPO/KTO baselines reported in Table 2 are trained on Human Preference Dataset as in their original setting.** DiffusionDPO and DiffusionKTO in their orginal paper's setting are designed to train on human preference dataset. Our method are able to train even without human preference dataset. However, to ensure fair comparsion and to evaluate our method in their set up, we also report Table 2, where DiffusionDPO and DiffusionKTO are trained on human preference dataset(pick-a-pick dataset) while our method is also trained on this dataset but only using the wining sample.

---

> ### Author Response · Authors · 2025-11-21
> **(2/2) Response to Reviewer PZtZ**
>
> > W1: Unconvincing Baseline Performance, HPS score being low might suggest dubious experiments setup.
>
>
> **R1:** We argue that this is not due to a flawed setup, but is due to the fact that
> * *we construct image preference pair instead of using human annotated preference data. Under such setting, DiffusionDPO seems to be much more sensitive to how preference pair is constructed compared to DiffusionKTO.*
>
> Below we illustrate this fact by restating the result from **Table 6 in the appendix** material(we show result from HPS dataset here):
> | Method             | PS    | CLIP  | HPS   | IR    |
> |--------------------|-------|-------|-------|-------|
> | Diff-DPO-sd35      | 79.50 | 49.00 | 76.50 | 74.50 |
> | **Diff-DPO-SFTsd15**   | **77.00** | **54.75** | **59.00** | **70.00** |
> | Diff-DPO-quality   | 75.75 | 56.00 | 54.75 | 68.00 |
> | Ours-TDPO          | 83.25 | 56.00 | 79.00 | 82.25 |
>
>
> | Method             | PS    | CLIP  | HPS   | IR    |
> |--------------------|-------|-------|-------|-------|
> | Diff-KTO-sd35      | 77.25 | 52.25 | 72.25 | 75.00 |
> | **Diff-KTO-SFTsd15**   | **80.25** | **53.75** | **76.00** | **76.75** |
> | Diff-KTO-quality   | 80.75 | 52.75 | 76.50 | 75.50 |
> | Ours-TKTO          | 80.00 | 55.75 | 81.00 | 80.75 |
>
> As one can see from the table, CLIP and HPS metrics from DiffusionDPO are quite sensitive to different ways of preference image construction, while DiffusionKTO is relatively stable. For a balanced performance between Diff-DPO and Diff-KTO we reported Diff-DPO-SFTsd15 and Diff-KTO-SFTsd15 in Table 1.
>
> We would also like to note that previous work has also found that DiffusionDPO can be unstable and struggles on the HPS metric even when trained on a human preference dataset. As shown in DSPO[1] Table 1, they witnessed that the DiffusionDPO baseline has a significant performance drop on the HPS score compared to the Supervised Fine-Tuning Baseline.
>
> >W2: Potentially Unfair Comparison, the paper should include an equivalent synthetic-image baseline for a properly controlled comparison.
>
> **R2:** As clarified in **C.1**, the baselines shown in Table 1 are from a synthetic image baseline (Figure 4.c). Where losing image for diffusionDPO/KTO is generated from the same losing prompt used by our method.
>
> Morever, our Table 2 shows baselines of DiffusionDPO in it's original setting, trained on a human preference dataset, while our method trained on the same dataset using only the wining image. This is a even more fair setting consider DiffusionDPO have access to additional access to additional access to human dis-prefered image while our method doesn't, we restate the result as below.
> | Method      | Supervision        | PS    | CLIP  | HPS   | IR    |
> |------------|--------------------|-------|-------|-------|-------|
> | Diff.-DPO* | (x_w, x_l, c)      | 71.60 | 58.80 | 70.20 | 63.60 |
> | Ours-TDPO  | (x, c_w, c_l)      | 70.60 | 62.60 | 75.80 | 75.80 |
>
>
> >W3: Opaque Training Process: No training dynamics, cruves provided. This makes it impossible to assess if the models are learning the intended preferences or if the training is stable.
>
> **R3**  We appreciate for your suggestions on showing the training dynamics. We have updated the paper to include training curves in Appendix Section A.8 (paragraph *Training Curves of TDPO and TKTO*). These curves show that the implicit preference score and evaluation metrics increases steadily, indicating that the model is learning the underlying preferences as expected.
>
> [1] https://openreview.net/pdf?id=xyfb9HHvMe

---

> ### Author Response · Authors · 2025-11-29
>
> We thank the reviewer again for taking the time to review our paper and for providing thoughtful and constructive feedback. As the discussion phase is coming to an end, we would like to kindly ask whether our previous clarifications have helped address the reviewer’s concerns—particularly regarding the fairness of our experimental setup and the other points raised. If there are remaining issues that require further clarification or adjustment, we would be grateful for any additional guidance on how we might improve the manuscript.

---

### Author Response · Authors · 2025-11-21

## Global response:
We thank all reviewers for their constructive feedback and for recognizing the novelty of aligning diffusion-based T2I models using automatically constructed text-level preferences (PZtZ, fwFA, 6UbU). We are also encouraged that reviewers found our perspective on “free-lunch” scalable alignment and our model-agnostic TDPO/TKTO framework both interesting and practically valuable (PZtZ, 6UbU). In addition, we appreciate the positive feedback on the strength of our experimental evaluation across multiple datasets, metrics, and ablations, which reviewers described as solid and convincing (fwFA, 6UbU).
We resolve all the concerns and replied to each reviewer saperately

### List of updates on the revised version of paper
* Quantative result of human studies comparing TDPO/DiffusionDPO/SFT:  We add a new paragraph *Quantitative results from Human Studies* in Section 4.2, a new Appendix section A.7 *User Study*, a new Winrate plot Figure 6 and a new User study interface Figure 12.
* Training Curves and dynamic: We add a new Appendix section A.8 *Training Curves of TKTO and TDPO*, and a new figure, Figure 13.
* Additional result on SDXL: Appendix B.4
* Additional result on choice of modifying LLMs: Appendix B.5

---

### Meta-Review · Area_Chair_5HzJ · 2026-01-05

**Summary:**

This paper proposes a method, called text preference optimization (TPO), for aligning T2I diffusion models without requiring human-annotated image preference pairs. It uses a LLM to generate text prompts that do not match the original caption, thus, creating text preference pairs (accepted/rejected), and then uses them with LLM-based alignment algorithms, such as DPO and KTO.

- The idea of starting without preference data and then using a LLM to generate preferred and dispreferred text prompts is very simple. However, the method would be biased towards the LLM generating matched/mismatched text prompts and its overall performance would heavily depend on the quality of the LLM. I also believe it is too strong to call such a method "free lunch alignment". The reviewers have the same concern about the effect of the LLM and view the work more like an engineering simplification than a fundamentally new learning principle.

- The reviewers also have major concerns regarding the experimental setup and results.

- Finally, the idea is not novel either. A similar approach was used in "Confidence-aware Reward Optimization for Finetuning Text-to-Image Models (ICLR-2024)" to generate prompts that are synthetically similar but semantically different than the original caption. Of course, the resulting data was used to calibrate different reward scores to address reward hacking in T2I RLHF.

As a result, I do not think the paper carries enough novelty and requires major improvements.

**Reviewer Concerns:**

I do not think the main concerns of the reviewers, especially Reviewers PZtZ and 6UbU, were addressed, because they are mainly related to the lack of novelty in the main contributions of the paper.

**Reviewer Scores:**

Unfortunately, none of the reviewers reacted to the authors' response. However, I doubt any of them would have changed their score, because their main concerns are related to the simplicity and the lack of novelty in the main contributions of the paper.

---

### Decision · Program_Chairs · 2026-01-26

Reject